



# Development and characterization of the Portable Ice Nucleation Chamber 2 (PINCii)

Dimitri Castarède[1,*], Zoé Brasseur[2,*], Yusheng Wu[2], Zamin A. Kanji[3], Markus Hartmann[1],
Lauri Ahonen[2], Merete Bilde[4], Markku Kulmala[2], Tuukka Petäjä[2], Jan B. C. Pettersson[1], Berko Sierau[5],
Olaf Stetzer[b], Frank Stratmann[6], Birgitta Svenningsson[7], Erik Swietlicki[7], Quynh Thu Nguyen[4,a],
Jonathan Duplissy[2,8], and Erik S. Thomson[1]

[1]Department of Chemistry and Molecular Biology, Atmospheric Science, University of Gothenburg, Gothenburg, Sweden
[2]Institute for Atmospheric and Earth System Research/Physics, Faculty of Science, University of Helsinki, Helsinki, Finland
[3]Institute for Atmospheric and Climate Science, ETH Zürich, Zürich, Switzerland
[4]Department of Chemistry, Aarhus University, Aarhus, Denmark
[5]Department of Health and Environment, City of Zürich, Switzerland
[6]Leibniz Institute for Tropospheric Research, Leipzig, Germany
[7]Department of Physics, Division of Nuclear Physics, Lund University, Lund, Sweden
[8]Helsinki Institute of Physics, University of Helsinki, Helsinki, Finland
[a]now at: Danish Technological Institute, Aarhus, Denmark
[b]previously at: Institute for Atmospheric and Climate Science, ETH Zürich, Zürich, Switzerland
[*]These authors contributed equally to this work.

**Correspondence:** Erik S. Thomson (erik.thomson@chem.gu.se) and Zoé Brasseur (zoe.brasseur@helsinki.fi)

**Abstract.** The Portable Ice Nucleation Chamber 2 (PINCii) is a newly developed continuous flow diffusion chamber (CFDC) for measuring ice nucleating particles (INPs). PINCii is a vertically-oriented parallel plate CFDC that has been engineered to improve upon limitations of previous generations of portable CFDCs. This work presents a detailed description of the PINCii instrument and the upgrades that make it unique compared to other operational CFDCs. The PINCii design offers
several possibilities for improved INP measurements. Notably, a specific icing procedure results in low background particle counts, which demonstrates the potential for PINCii to measure INPs in very low concentrations. High spatial resolution wall-temperature mapping enables the identification of temperature inhomogeneities on the chamber walls. This feature is used to introduce and discuss a new method to analyze CFDC data. A temperature gradient can be maintained throughout the evaporation section in addition to the main chamber, which enables PINCii to be used to study droplet activation processes or to extend ice crystal growth. A series of both liquid droplet activation and ice nucleation experiments were conducted at
temperature and saturation conditions that span the spectrum of PINCii's operational conditions ($-50 \leq$ temperature $\leq -15\,°\mathrm{C}$ and $100 \leq$ relative humidity with respect to ice $\leq 160\,\%$) to demonstrate the capabilities of PINCii. In addition, typical sources of uncertainty in CFDCs, including particle background, particle loss, and variations in aerosol lamina temperature and relative humidity, are quantified and discussed for PINCii.



## 1 Introduction

Ice crystals are abundant in the atmosphere, and approximately 70 % of all cloud occurrences contain ice (Stubenrauch et al., 2013; Matus and L'Ecuyer, 2017). Yet, ice crystal formation and multiplication processes are poorly represented in atmospheric models (Fletcher, 1962; Phillips et al., 2008; Niemand et al., 2012; Burrows et al., 2022; Frostenberg et al., 2022). In the atmosphere, ice crystals can form from supercooled aqueous droplets via *homogeneous ice nucleation*, however, this process

occurs only at temperatures $T \lesssim -37°C$ (Koop et al., 2000b; Murray et al., 2010). In fact, most atmospheric ice results from crystallization processes that occur at higher temperatures ($-37°C < T < 0°C$). This process, called *heterogeneous nucleation*, requires the presence of aerosol particles (Hoose and Möhler, 2012). Heterogeneous ice nucleation may occur within existing water droplets (immersion freezing, contact freezing, pore condensation and freezing; Ladino Moreno et al., 2013; Vali et al., 2015; David et al., 2019) or at sub-saturated water conditions directly from the vapor phase (deposition nucleation; Marcolli,

2014; Vali et al., 2015).The multiple mechanisms make understanding atmospheric ice nucleating particles (INPs) and their roles challenging (Vali et al., 2015).

The desire to better understand INPs has led to the development of different instruments and methods for investigating atmospheric ice nucleation mechanisms and processes since the 1940's. These instruments include cloud chambers (Aufm Kampe and Weickmann, 1951; Mason, 1962; DeMott et al., 2011; Möhler et al., 2021), filter sampling and nucleation testing methods

(Langer and Rodgers, 1975; Conen et al., 2012; Hill et al., 2014; Stopelli et al., 2014; Schrod et al., 2016; Schiebel, 2017; Chen et al., 2018; Porter et al., 2020), and flow reactors, including continuous flow diffusion chambers (CFDCs). Since they were first developed in the 1980s, multiple generations of CFDCs have emerged for making semi-continuous, online ambient measurements of INPs (Hussain and Saunders, 1984; Tomlinson and Fukuta, 1985; Rogers, 1988; Petters et al., 2009; Richardson et al., 2007; Richardson, 2009; Kanji and Abbatt, 2009; Eidhammer et al., 2010; Richardson et al., 2010). In the

mid-2000's, a vertical parallel plate CFDC, called the Zurich Ice Nucleation Chamber (ZINC), was introduced (Stetzer et al., 2008). After the initial success of this chamber in laboratory studies, more parallel plate CFDCs were constructed. A Portable Ice Nucleation Chamber (PINC) was developed at ETH-Zurich with a focus on field and mobile measurements (Chou, 2011; Kanji et al., 2013, 2019). Several commercialized vertical CFDCs, such as the Droplet Measurement Technology SPIN chamber (Garimella et al., 2016) and the CFDC-IAS (Handix Scientific, Boulder, Colorado, USA), have also been produced, refined,

and made available over the last decade. Horizontal parallel plate CFDCs were also developed (Kanji and Abbatt, 2009; Lacher et al., 2017). For instance, the automated Horizontal Ice Nucleation Chamber (HINC-Auto) is the first fully automated CFDC, and has operated continuously at the Jungfraujoch for more than a year (Brunner and Kanji, 2021).

Despite the number of CFDCs that have been developed, operated, and continue operation today, new systems continue to appear, and to-date, no instrument standard has emerged. Here we present a new instrument that has been built through a

large multi-national research collaboration (Boy et al., 2019) in an attempt to address specific short-comings of many existing instruments. This instrument, the second generation Portable Ice Nucleation Chamber (PINCii), incorporates unique hardware and operational improvements. For example, engineering upgrades have been made to the ZINC/PINC based design, and the temperature control in PINCii has been improved by the addition and better distribution of thermocouples and coolant





injection(s). The minimum wall temperature achievable with PINCii is $\sim -67°$C due to its cascade cooling system. In addition,
we show that the entire chamber can be run with a temperature gradient (including the evaporation section), meaning that, in
addition to its ice nucleation mode, the chamber has the potential to extend ice crystal growth and to act as a low-temperature
Cloud Condensation Nuclei Counter (CCNC). Finally, during PINCii's characterization and evaluation, new interpretation
methods for CFDC observations have been developed to better account for temperature and relative humidity (RH) uncertainties
within the chamber.

## 2    Instrument design and operation

CFDCs follow a common principle, whereby a thin layer of sample air containing particles, hereby called a "lamina", is injected
into a chamber (Rogers, 1988). The lamina is sandwiched between two layers of dry and clean sheath air, and the total volume
of sample plus sheath air flows in a laminar regime through an ice-coated chamber(s) that is held at sub-zero temperatures.
Most CFDCs are composed of thermally separated sections with a "growth section" (or "main chamber") and an "evaporation
section", as shown in Fig. 1. PINCii operates by the same principle, and within the main chamber and the evaporation section,
a continuous ice coating acts as a source of water vapor because the ice slowly sublimates into the sample flow in order to
maintain the vapor pressure (p) at the equilibrium vapor pressure of ice ($p_{i,eq}$), $p = p_{i,eq}(T)$. In static isothermal conditions, the
chamber walls equilibrate with the flow at ice saturation $RH_i(T) = 100$ %. When a temperature gradient is applied by setting
the temperatures of the walls to different setpoints, the resulting linear temperature and vapor pressure gradients lead to an
actual vapor pressure away from the walls that exceeds the equilibrium vapor pressure of ice (i.e., $p > p_{i,eq}$). This is due to the
fact that there is an exponential relationship between temperature and saturation vapor pressure (Clapeyron, 1834; Clausius,
1850; Rogers, 1988). Therefore, the supersaturated conditions with respect to ice and/or water enables ice nucleation or droplet
activation.

Sampled particles, which are travelling within the thin, sandwiched lamina, are exposed to only a narrow range of lamina
temperature ($T_{lam}$) and lamina relative humidity conditions ($RH_{lam}$) that depend upon the chosen set-point temperatures. For
a given $T_{lam}$ and $RH_{lam}$, particles may or may not nucleate ice depending on their capacity to act as INPs. Sampled particles
that are prone to act as INPs at the given lamina conditions nucleate ice and continue to grow in the main chamber section.
Meanwhile, the evaporation section is held in an isothermal condition at the temperature of the warmer main chamber wall.
This makes the evaporation section saturated with respect to ice (relative humidity with respect to ice $RH_i = 100$ %), but sub-
saturated with respect to liquid water (relative humidity with respect to water $RH_{liq} < 100$ %). Thus, any liquid droplets are
evaporated while ice particles remain at equilibrium. The ice crystals can ultimately be differentiated from other particles and
counted at the exit of the chamber using particle counting techniques.

### 2.1    PINCii design

PINCii is a 112 cm $\times$ 70 cm $\times$ 190 cm instrument that weighs approximately 200 kg and is mounted on wheels to be mobile. As
shown in Fig 1, the main chamber is 1 m long and 33 cm wide and is constructed of two parallel aluminum walls, which were





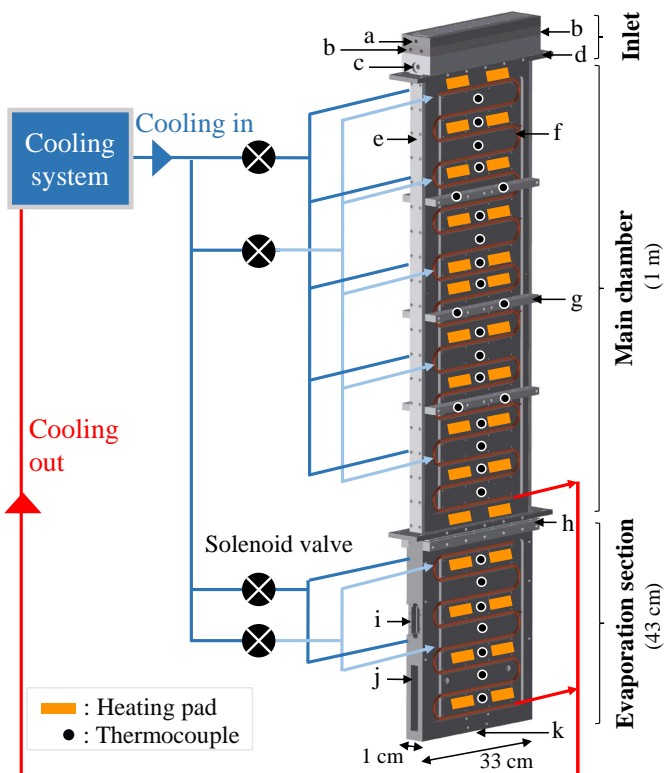

**Figure 1.** Schematic of PINCii and its cooling system, including the main elements of the chamber: (a) the sample inlet, (b) the sheath flow inlets, (c) the water level sensor port, (d), (e) and (h) the SustaPEEK flanges thermally isolating sections of the chamber, (f) refrigerant cooling coil pipes, (g) support bars, (i) window port, (j) the lower evaporation section with material removed, and (k) the exit hole. The location of the coolant injections, heating pads and thermocouples on a single wall are also depicted. The injection of coolant to the different chamber sections is controlled independently, while multiple thin capillaries located after the solenoid valves distribute the coolant evenly to the selected section.

sandblasted and anodized to a depth of $20 \pm 4$ µm. The aluminum walls are separated by 1 cm of semi-crystalline thermoplastic (SustaPEEK) sidewall pieces (Fig. 1e). The SustaPEEK material was chosen for its material properties after low-temperature experiments with polyvinylidene fluoride proved problematic due to mismatched cooling properties that led to chamber leakage at low temperatures. Flanges of SustaPEEK are also used to thermally isolate individual sections from others (Fig. 1 d and h).

An aerosol inlet head unit is located on top of the main chamber and composed of four machined pieces to create two sheath flows sandwiching the sample lamina. The inlet head unit has 6 ports, the sample flow inlet (Fig. 1a), four sheath flow inlets (2 per side, Fig. 1b), and a water level sensor port (Fig. 1c). After entering, the sample flow is channeled through a 1 mm slit sandwiched with the sheath flows on each side. Located below the main chamber, the evaporation section is 43 cm long and contains a two-sided window port (Fig. 1i) designed for mounting the ETH-IODE polarization detector (Nicolet et al., 2010).

After 20 cm, the 33 cm width of the evaporation section narrows smoothly to a 1 cm exit hole (Fig. 1k) where a chosen detector,





such as an Optical Particle Counter (OPC), can be mounted. To lighten the unit and optimize temperature control, material has been removed where the lower evaporation section begins (Fig. 1j). Unlike the main chamber that has thermally isolated walls, the evaporation section is constructed of two connected aluminum pieces. The purpose of the thermal connection is to allow the evaporation section to operate efficiently in an isothermal mode. However, the evaporation section can also act as an elongation of the main chamber by extending the temperature gradient along its walls. Figure 1 also highlights the refrigerant cooling coil pipes (Fig. 1f) that are press-fit into external, machined grooves on the chamber walls and the support bars (Fig. 1g) that are mounted to reduce chamber distortions from material contraction and/or expansion during cooling and heating cycles.

The PINCii has a two-stage cascade refrigerant compressor system that provides independent cooling to the two upper chamber walls and the two evaporation section walls via four solenoid valves (Fig. 1). To distribute cooling fluid evenly, a series of thin injection capillaries are distributed across each wall after the solenoid valves. The design of this system was optimized to reduce singular, cold injection points given that the R23 coolant operates with an evaporation temperature of -72 °C. With the instrument design and heat losses, the actual observed minimum wall temperature is $\sim -67°$ C. In addition to the cooling system, 60 adhesive silicon heating pads (output power: 20 W each) and 56 type-K thermocouples measuring at 1 Hz (embedded $\sim 7$ mm into the chamber walls, or $\sim 5$ mm from the ice layers) are used to control the temperature via 11 individual PID loops controlled with a custom-made LabVIEW program. The same program is used to operate a series of external pumps, mass flow controllers (MFCs) and valves used to manage the flows inside the chamber, and to record all the output data, including the OPC data.

## 2.2 Features unique to PINCii

Given the abundance of operational portable CFDCs (Chou, 2011; Garimella et al., 2016; Lacher et al., 2017; Brunner and Kanji, 2021), it is useful to distinguish unique features of the PINCii design. Some of these new design features initially motivated the PINCii project, while others have emerged during the development process. In fact, the chamber developed and introduced here is PINCii version 3.0, and to date three PINCii chambers of this version have been commissioned. While further incremental improvements are likely to emerge, it is likely they will do so in the context of the user community. The version presented here is considered to be the reproducible PINCii package for which the hardware and operation can be easily reproduced.

Compared to other deployable CFDCs, PINCii has an elongated design, where the main chamber and the evaporation section are longer than in other instruments. The extended chamber(s) enable longer residence time(s) compared to other existing chambers and thus more time for nucleation and growth of both ice particles and water droplets in the main chamber and more time for water droplet evaporation in the evaporation section. As mentioned previously, the evaporation section can also act as an elongation of the main chamber by extending the temperature gradient applied to the main chamber. This feature can be used to perform droplet nucleation experiments or to expand ice crystal growth below water saturation where phase differentiation is not critical, particularly for low temperature ice nucleation experiments where growth kinetics are limiting.

Furthermore, the capillary distribution of the coolant throughout the chamber, together with a dense spatial distribution of thermocouples and heating pads and a temperature control system, was developed to reduce local temperature variability in



the walls. The dense temperature monitoring has also enabled improved analysis and we suggest new protocols to analyze and interpret CFDC data in section 3.3.

## 2.3   Typical chamber operation

In typical PINCii experiments, the chamber walls are coated with $\sim 1$ mm thick ice layers to provide the water vapor source within the chamber. The icing procedure consists of cooling the walls and flushing the chamber with water (more details

in Section 4.1). As previously outlined, the driving force for ice nucleation is established by setting a temperature gradient between the walls. As a standard, a 1 L·min$^{-1}$ sample aerosol flow is sandwiched between two 4.5 L·min$^{-1}$ dry nitrogen or dry/clean air sheath flows, resulting in a total flow of 10 L·min$^{-1}$ through the chamber. The flow rates have been chosen to constrain the sample within a lamina with well defined $T_{lam}$ and $RH_{lam}$ and to avoid wall effects (Fig. 4). A thermodynamic model that includes the attenuating effect of the ice layer is described in more detail below (cf. §2.4).

All the experiments presented in this study were conducted using a four-channel OPC (Remote 3104, Lighthouse worldwide solutions, USA) mounted at the exit of PINCii. The size channels (Ch) correspond to the following sizing bins, with $d_p$ the optical particle diameter ; Ch1: $0.3 \leq d_p < 1$ μm; Ch2: $1 \leq d_p < 3$ μm; Ch3: $3 \leq d_p < 5$ μm; Ch4: $5 \leq d_p < 25$ μm. In some of the results presented here, the channels have been cumulatively combined to result into Ch1: $d_p \geq 0.3$ μm; Ch2: $d_p \geq 1$ μm; Ch3: $d_p \geq 3$ μm; Ch4: $d_p \geq 5$ μm. Note that the OPC was recalibrated to operate with a 10 L·min$^{-1}$ flow rate.

## 2.4   Thermodynamic model

The actual thermodynamic conditions to which the sample is exposed are numerically modeled using the thermodynamic model of Rogers (1988) that has served as the reference for all subsequent CFDCs. The model quantifies temperature, RH, and flow fields within the chamber according to the measured flow rates and wall temperatures (Fig. 2). Note that the calculations detailed by Rogers (1988) use several approximations which could potentially be adjusted depending on the CFDC and the

experimental conditions used. For example, Patnaude et al. (2021) modified the approximations to better represent the geometry of their cylindrical CFDC and measurements at low temperatures and flow rates. Here we updated the model to account for the 1 mm thick ice layers on the side walls (hatched gray region, Fig. 2). This thickness value is calculated by measuring the total volume of water exiting the chamber after an icing / melting cycle, and dividing it by the total surface of the chamber walls. Therefore, the machined 1 cm gap between the chamber walls becomes 0.8 cm and the total chamber volume is decreased by

$\sim 20$ %. The chamber flow and thermodynamic conditions presented in Fig. 2 are calculated across these two ice layers whose temperatures are calculated accounting for the heat transfer between the aluminum chamber walls (in which the thermocouples are embedded) and the ice/gas interface. It is noteworthy that accounting for the ice layer thickness changes the $T_{lam}$ and $RH_{lam}$ and simultaneously changes the lamina position and the velocity profile across the chamber (Figs. 2 and 3). Furthermore, while the standard model predicts flow reversal due to buoyancy, as shown in Garimella et al. (2016) and here in Fig. 3 with the

purple negative velocity region, the updated model predicts a laminar flow profile with negligible reversal even when strong temperature gradients are applied between the chamber walls, as seen in Fig. 3 as well as in Fig. 4, where the fraction of reversed flow is represented over the full range of $T_{lam}$ and $RH_{lam}$ achievable with PINCii.



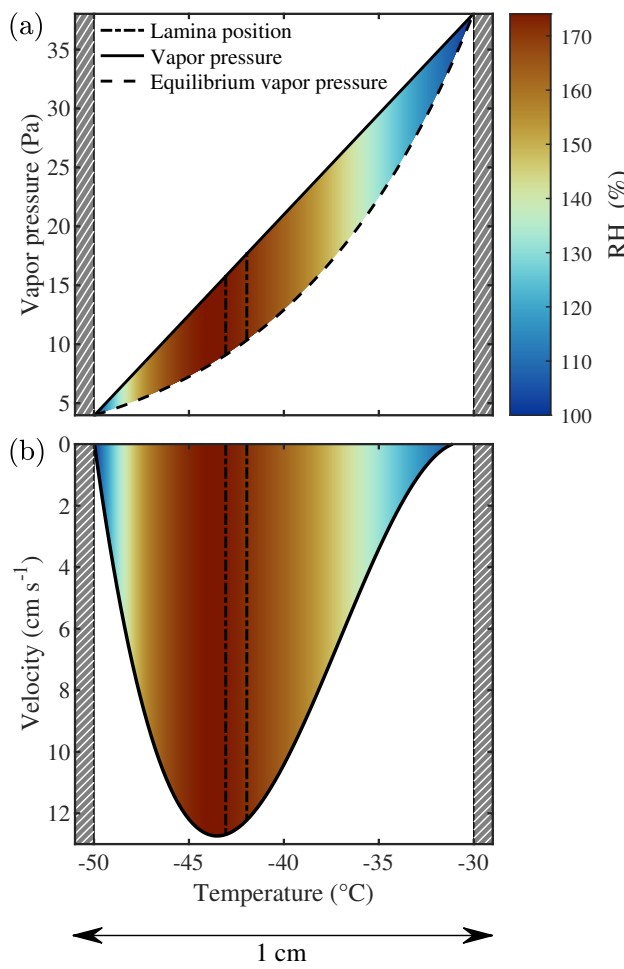

**Figure 2.** Thermodynamic model output for the PINCii main chamber when the cold wall (left) is fixed at T = -50 °C and the warm wall (right) at T = -30 °C. In the top panel (a), the linear gradient of vapor pressure (solid line) is plotted together with the theoretical equilibrium vapor pressure over ice (dashed line). The bottom panel (b) displays the flow velocity profile (solid line). In both panels, the sample lamina position (within the dash-dotted lines), the predicted $RH_i$ (colormap) and the 1 mm thick ice layers (hatched gray region) are depicted.





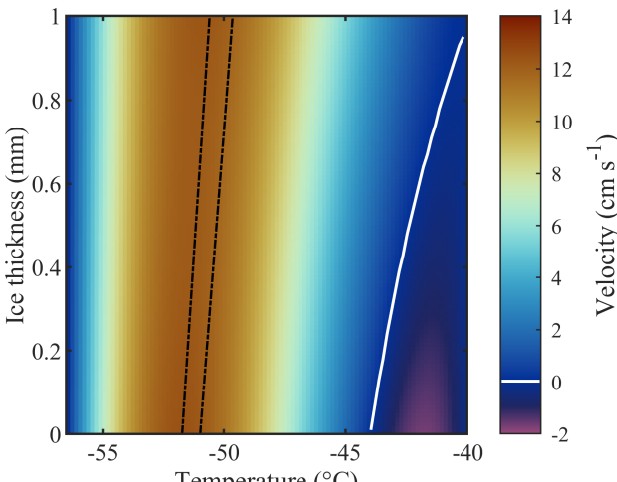

**Figure 3.** Flow velocity as a function of ice thickness for fixed wall temperatures of -40.0 and -56.5 °C, chosen to represent homogeneous freezing conditions in PINCii's main chamber. These conditions ($T_{lam}$ = -51.3 °C and $RH_{i,lam}$ = 155.7 % at ice thickness = 0 mm & $T_{lam}$ = -50.1 °C and $RH_{i,lam}$ = 154.7 % at ice thickness = 1 mm) are representative of extreme chamber operations for PINCii, with the greatest potential for buoyancy effects. The lamina position is depicted by the dashed black lines. The white contour line corresponds to a velocity of 0 cm/s and emphasizes where the region with negative velocity starts.



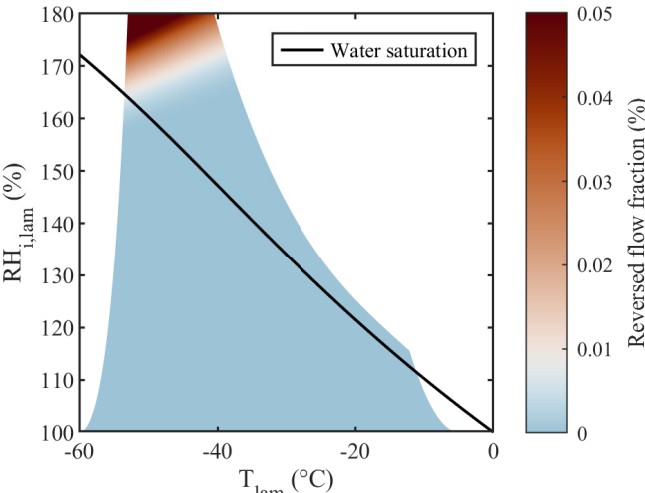

**Figure 4.** Achievable $T_{lam}$ and $RH_{i,lam}$ assuming fixed wall temperatures between -5 and -60 °C and accounting for the droplet breakthrough results presented in section 3.1. The color map represents the reversed flow fraction defined as the ratio between the reverse (upward) flow and the normal (downward) flow in the chamber, assuming a 1 mm ice layer on each wall. The black solid line represents water saturation ($RH_{liq,lam}$=100 %).

## 3 Evaluating the chamber performance

The primary goal of this paper is to present the ability of PINCii to operate reliably and precisely at conditions relevant for
atmospheric ice nucleation. Thus, the chamber was characterized and evaluated using a series of experiments that induce observable activation and/or nucleation processes and span the range of typical operating conditions. Droplet nucleation, droplet breakthrough, deliquescence, and homogeneous and heterogeneous freezing experiments have been performed and compared to theoretical predictions and previous experimental work (Köhler, 1936; Koop et al., 2000b, a; Welti et al., 2009).

### 3.1 Droplet nucleation and deliquescence

In addition to its use for ice nucleation experiments, PINCii has the potential to be operated in a low temperature CCNC mode whereby a temperature gradient is maintained along the entire chamber length, including along the evaporation section. This effectively extends the main chamber, allowing droplets to continue to grow. In this work, we show that the chamber can actively grow droplets. We study the activation of polydisperse ambient aerosol particles and the deliquescence of 200 nm Sodium Chloride (NaCl) particles. The objective of these experiments is to evaluate PINCii's performance by comparing the
$RH_{i,lam}$ measured to values predicted by theory and previous laboratory experiments, both for the deliquescence of NaCl and for water saturation ($RH_{liq}$=100 %). Further characterisation to evaluate PINCii's use as a CCNC would require experiments





with well characterized salts of known size distribution and chemical composition, which we do not explore further in this work.

Results from the droplet nucleation experiments are presented in Fig. 5. Two experiments were conducted with ambient
laboratory air (i.e. polydisperse particles), where we expect larger particles (in the order of magnitude of a micrometer) to activate just above water saturation (Köhler, 1936). The experiments were conducted on two separate days, and because the concentration and composition of particles from the ambient laboratory air may have changed from one day to another, the results presented in Fig. 5 were normalized by the maximum droplet concentration reached during each ramp. During the experiments, the $RH_{liq,lam}$ was increased from 90 to 110 % at a rate of 1 %·min$^{-1}$, and in total seven ramps were conducted
with $T_{lam}$ between -20 and -35 °C. The temperature was kept above -37 °C because at colder temperatures homogeneous freezing processes obscure observations of droplet formation. For each ramp, droplet activation was observed at $RH_{liq,lam}$ ≈ 100 % as predicted by the Köhler theory for the activation onset of microscopic particles (Köhler, 1936; Castarède and Thomson, 2018). At $RH_{liq,lam}$ > 100 %, the continually increasing concentrations of hydrometeors with $d_p \geq 1\mu m$ may either represent the activation of a larger fraction of the aerosol population as supersaturation allows small particles to overcome the
Kelvin barrier (Thomson, 1872; Orr et al., 1958), or the immersion freezing of nucleated droplets if INPs are present within the sampled particles.

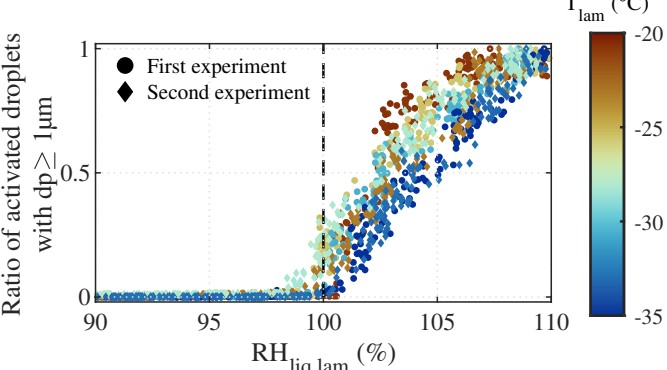

**Figure 5.** Normalized concentration of droplets larger than 1 μm as a function of the $RH_{liq,\,lam}$ and temperature during the two droplet activation experiments. RH ramps were conducted from $RH_{liq,lam}$ = 90 to 110 % at a ramping rate of 1 %·min$^{-1}$. Droplet formation was observed at $RH_{liq,lam}$ ≈ 100 % (dashed line).

Experiments with deliquescing NaCl were performed to establish PINCii's coherence with theory well below water satura-
tion. Deliquescence represents the transition from a soluble particle to a saturated solution droplet (Hämeri et al., 2001; Cheng et al., 2015; Castarède and Thomson, 2018) and is predicted using Köhler theory, which accounts for the size and compo-
sition of the initial particulate and resulting brine droplet (Köhler, 1936; Castarède and Thomson, 2018). The deliquescence experiments were conducted with NaCl particles, which were generated with an atomizer, dried and then size selected at 200 nm using an Electrostatic Classifier and Differential Mobility Analyzer (DMA) (TSI models 3080 and 3081 respectively). The



nm NaCl particles were then delivered continuously to PINCii while increasing the $RH_{liq,lam}$ gradually from 70 to 105 %. Results from the experiment conducted at -35 °C are presented in Fig. 6 and compared to experimental deliquescence data

obtained at cold temperature by Koop et al. (2000a). Note that, to account for any variation in the concentration of particles injected in the chamber, the results are presented as the ratio between the concentration of particles measured by the OPC at the exit of PINCii and the total particle concentration measured using a condensation particle counter (CPC; TSI model 3775) mounted in parallel to PINCii's inlet. The deliquescence of the 200 nm NaCl particles is difficult to observe with this setup due to the coarse resolution of the OPC and given that the growth factor for NaCl particles is $\sim 1.8$ when transitioning from

their dry to solvated state (Hämeri et al., 2001; Biskos et al., 2006; Cheng et al., 2015). The size transition is however visible in the smallest OPC size channel, $0.3 \leq d_p < 1$ µm, where we observe an increase in the concentration ratio at $RH_{liq,lam} \approx 77$ %, which agrees well with the earlier observations from Koop et al. (2000a) (vertical dashed line in Fig. 6). Although these results clearly show PINCii's potential to study droplet activation, its use as a low temperature CCNC would require a more rigorous implementation and characterization. Among other things, a higher resolution optical detector should be used at the exit of the

chamber, and more experiments should be conducted with well characterized salts.

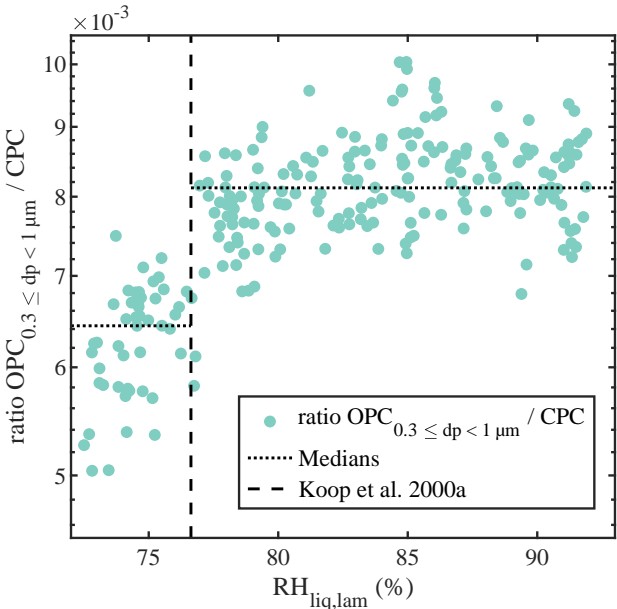

**Figure 6.** Concentration of dry/solvated NaCl particles with $0.3 \leq d_p < 1$ µm divided by the particle concentration measured with the CPC mounted in parallel to PINCii's inlet, as a function of $RH_{liq,lam}$ at $T_{lam} = -35$ °C. Deliquescence is observed where the concentration increases at $RH_{liq,lam} \approx 77$ %, which is the deliquescence relative humidity also reported by Koop et al. (2000a) and represented here with the vertical dashed line. To emphasizes the deliquescence, the median values of the concentration ratio before (6.4e−3) and after (8.1e−3) the deliquescence relative humidity are shown as dotted lines.

## 3.2 Droplet breakthrough

Droplet breakthrough refers to chamber conditions at which the evaporation section no longer functions effectively. In these conditions, liquid water droplets grow large enough in the main chamber that they can propagate through the evaporation section and into the optical detector. This results in large droplets being counted in the same size channels as ice crystals, and thus particle phase can no longer be reliably determined. There exist some phase discriminating detectors where phase can be directly determined during detection (Nicolet et al., 2010; Garimella et al., 2016; Mahrt et al., 2019), but these have other experimental challenges limiting their use. Moreover, for the purposes of a general introduction, the use of the simplified evaporation section-to-OPC coupling is considered to be best practice. It is with such a system that results between chambers can be most straightforwardly discussed, and also where quantifying droplet breakthrough becomes important.

In Fig. 7, results from droplet breakthrough experiments are shown. These experiments were used to characterize the upper limit of the operating conditions for which we can achieve reliable measurements of ice crystals without the impact of droplet breakthrough. The experiments consisted of ramps from $RH_{liq,lam}$=95 to 115 % for four selected $T_{lam}$, -20, -25, -30 and -35 °C, with an aerosol sample consisting of 200 nm ammonium nitrate ($NH_4NO_3$) particles. These particles are not generally ice active, and thus deliquesce to form droplets and continue to grow with increasing RH (Köhler, 1936). Here, we conservatively define droplet breakthrough as the supersaturation point $RH_{DB}$ where the droplet concentration exceeds $1 \cdot cm^{-3}$ in the largest size-channel ($d_p \geq 5\mu m$; dashed line in Fig. 7). Droplet breakthrough was observed at $RH_{DB}$ of 103, 104.5, 107.5 and 114.2 % for the experiments at $T_{lam}$ of -20, -25, -30 and -35 °C respectively, indicating that PINCii's evaporation section allows for experiments to be performed at a few percent above water saturation, even at the warmest tested $T_{lam}$, without needing to differentiate droplets from ice crystals.

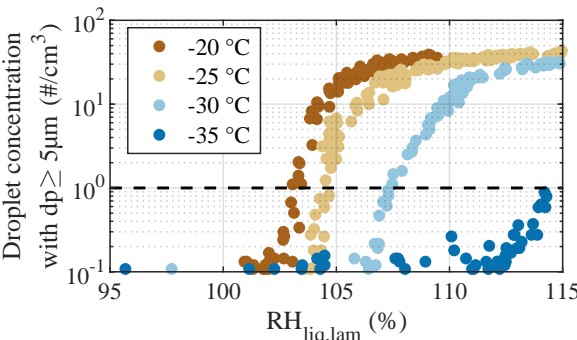

**Figure 7.** Concentration of droplets with dp≥5µm as a function of $RH_{liq,lam}$ during the droplet breakthrough experiments. RH ramps were conducted from 95 to 115 % at a rate of 1 %·min$^{-1}$ for four fixed $T_{lam}$ of -20, -25, -30 and -35 °C using 200 nm $NH_4NO_3$ particles. Droplet breakthrough is defined as occurring when the droplet concentration exceeds $1 \cdot cm^{-3}$ (dashed line).



### 3.3 Homogeneous and heterogeneous freezing experiments

#### 3.3.1 Homogeneous freezing experiments

To compare with Koop et al. (2000b), who model homogeneous freezing from solution droplets, experiments were performed using aerosolized 200 nm NaCl particles at -51 °C $\leq T_{lam} \leq$ -39 °C, and the results are presented in Fig. 8. The theoretical homogeneous freezing onset is plotted for a water-activity criterion $\Delta a_w = 0.2946$ (Koop et al., 2000b). This water-activity criterion is calculated assuming the homogeneous freezing of pure water at -36.45 °C, the warmest homogeneous freezing temperature reported in Murray et al. (2010), and used to represent homogeneous freezing onset, absent kinetic limitations. In the three panels of Fig. 8, the activated fraction (AF) was calculated as the ratio between the concentration of particles exceeding a certain size threshold measured with the OPC and the total particle concentration measured using a CPC (TSI model 3775) mounted in parallel to PINCii's inlet. In Fig. 8 (a), the observed AF in the largest OPC size channel ($d_p \geq 5$ µm) is plotted as a function of the average lamina conditions, $T_{lam}$ and $RH_{lam}$. Presenting data in this way shows pre-activation ($RH_{lam} < RH_{Koop}$) at $T_{lam} >$ -45 °C, and delayed activation ($RH_{lam} > RH_{Koop}$) at $T_{lam} <$ -50 °C, with respect to the Koop et al. (2000b) curve. These deviations are consistent with previous CFDCs observations by Garimella et al. (2016); Welti et al. (2020); Brunner and Kanji (2021) who attribute these deviations to either uncertainty in the supersaturation or time-dependent effects (aqueous aerosol does not reach equilibrium before freezing).

In Fig. 8 (b), the size threshold used to calculate the AF is changed from 5 µm to 3 µm to account for the fact that growth of ice crystals becomes kinetically limited as temperature decreases (Rogers and Yau, 1989; Welti et al., 2020) and that, as a result, a fraction of nucleated crystals do not have long enough residence times to grow to the sizes $\geq 5$ µm to be counted in the OPC's largest detection channel. This is confirmed by calculating the ice crystal growth by diffusion for spherical ice crystals as a function of temperature for the typical residence time ($t_{res} \approx 15$ s) in PINCii (see Fig. A3 in the appendix). Following Welti et al. (2020), we determine that ice crystals grow up to 2.74 µm at $T_{lam} =$ -51 °C and $RH_{i,lam} = 140$ when using a mass accommodation coefficient for ice $\alpha = 0.3$ and a initial seed particle diameter $d_p = 200$ nm. Thus, although the change in the size threshold does not affect the activation curves for $T_{lam} >$ -45 °C, it illustrates the existence of activated ice crystals that have not grown fully to 5 µm for $T_{lam} <$ -48 °C. Note that similar observations were made by Burkert-Kohn et al. (2017) when comparing two CFDCs using different size thresholds and by Brunner and Kanji (2021) where different size thresholds were used for different measurement regimes. Given that, in these cases, the activation onset is observed below water saturation, there is no risk that lowering the size threshold for counting ice would introduce droplet counting (cf. section 3.2). However, kinetic limitations are not sufficient to explain the observed deviations, as Figs. 8 (a) and 8 (b) still show ice activation at lower driving force than predicted by the theoretical homogeneous freezing onset (Koop et al., 2000b).

An alternative explanation for the observed deviations can be found by looking at the conditions triggering homogeneous freezing. Homogeneous freezing is an irreversible process where ice formation occurs well above the equilibrium vapor pressure. Moreover, when the right conditions for nucleation exist and ice formation is stimulated, it will quickly drive water vapor towards the ice phase due to the strong supersaturation. This suggests that the critical condition for ice formation in PINCii is the lamina condition that represents the strongest p, or in other words, the most extreme ice-triggering condition. The high





spatial resolution of PINCii's temperature monitoring (thermocouples in Fig. 1) enables us to identify chamber-wall temper-
ature anomalies and calculate lamina conditions in more detail. Ideally, a complete vector analysis would be done in order to
identify the lamina conditions that are most likely to trigger homogeneous freezing based on the coordinate distances between
$T_{lam}$/$RH_{i,lam}$ and $T_{Koop}$/$RH_{i,Koop}$. However, given that homogeneous freezing onset is only weakly dependent on temperature
relative to the saturation condition, here we simplify the problem by finding the greatest $RH_{i,lam}$ within the lamina. We do
this by calculating the lamina conditions for paired thermocouples that are embedded into opposite chamber walls (cf. Fig. 1)
and selecting the most extreme lamina conditions (greatest $RH_{i,lam}$) present within the chamber. When re-plotted against these
extreme $RH_{i,lam}$ conditions in Fig. 8 (c), the activation onset shifts to better overlay the Koop et al. (2000b) line for all studied
temperatures.

To validate this new method for analyzing homogeneous freezing data, it was also applied to a more complex natural
salt sample collected from a saline lake in the Qaidam basin (China). More details concerning the sample collection and
composition can be found in Kong et al. (2022). The sample was wet-dispersed and size-selected (200 nm), and when the
results were plotted against the greatest $RH_{i,lam}$ while using a size threshold dp $\geq$ 3 μm, the activation onset also matched the
Koop et al. (2000b) prediction for the entire temperature spectrum, as shown in Fig. A1.



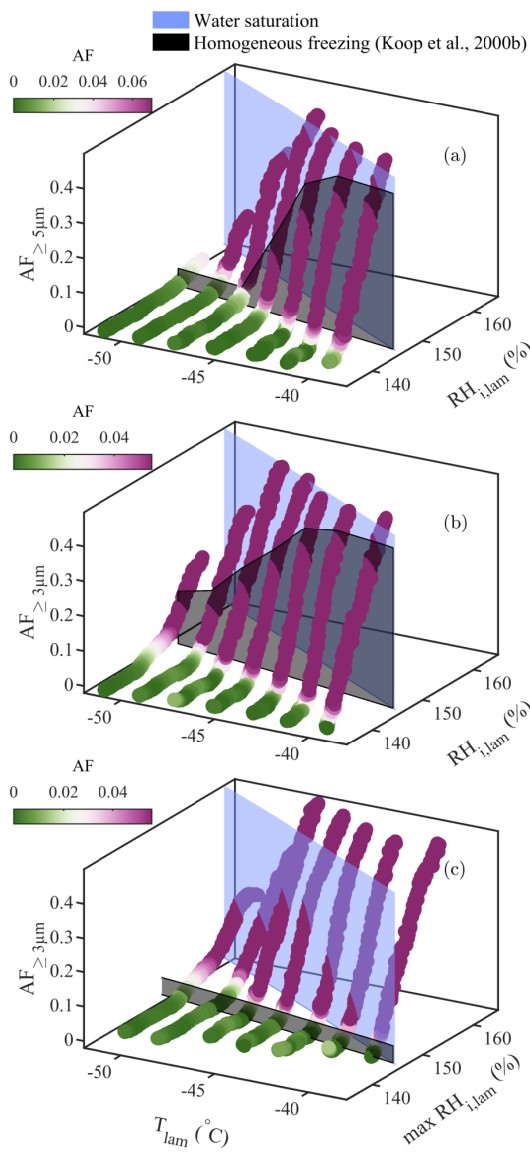

**Figure 8.** Homogeneous freezing of 200 nm NaCl particles plotted as AF as a function of $T_{lam}$ and $RH_{i,lam}$ represented in three different manners: (a) AF of aerosols with $d_p \geq 5$ μm plotted as a function of the average $T_{lam}$ and $RH_{i,lam}$. (b) AF of aerosols with $d_p \geq 3$ μm plotted as a function of the average $T_{lam}$ and $RH_{i,lam}$. (c) AF of aerosols with $d_p \geq 3$ μm plotted as a function of the average $T_{lam}$ and the maximum $RH_{i,lam}$. The 3d plot is used to emphasize the discontinuity associated with activation. The theoretical curves for water saturation (blue shading) and homogeneous freezing (black shading, $\Delta a_w = 0.2946$, Koop et al., 2000b) are projected into the 3-d space. Note that the extent of the black shading is limited and stops where it intersects with the measured AF. The black shading is represented in this way to allow the blue shading (water saturation) behind to remain visible. In each plot, the color scale is used to represent changes in the AF. The white region in the color bars represents the ice nucleation onset, which was estimated using the median of the inflection points obtained for each activation curve (see A1 for more information).





### 3.3.2 Heterogeneous freezing experiments

Heterogeneous freezing experiments are significantly more problematic to predict than homogeneous freezing and no robust
theory for *a priori* prediction is widely accepted. Even relying on past measurements can be problematic because uncon-
trolled variables, such as differences in experimental setups and material changes, can affect results significantly. That said,
certain materials such as minerals, like NX-illite, potassium feldspar (K-feldspar) and bacterial INP (e.g., Snomax®) have seen
widespread use in laboratory tests of heterogeneous ice nucleation and for instrument inter-comparison purposes (Welti et al.,
2009; Hiranuma et al., 2015; DeMott et al., 2018). Here we used NX-illite (Arginotec, NX Nanopowder; see Hiranuma et al.
(2015) for more information concerning the sample composition) from which particles were dry generated and size selected at
200 nm using the DMA. We conducted the experiments at fixed $T_{lam}$, and with $RH_{i,lam}$ ramped linearly from $RH_{i,lam}$ = 100 %
to $RH_{i,lam}$ = 150 % at 1 %·min$^{-1}$. In Fig. 9, the observed AF for particles with $d_p \geq 3$ µm is plotted and compared to experi-
mental observations from Welti et al. (2009) who performed similar experiments using the laboratory-based ZINC instrument
(Stetzer et al., 2008). Fig. 9 shows that our measured freezing onsets are in good agreement with the earlier measurements,
with the exception of the $T_{lam} \approx$ -35 °C scan, where we observe activation at a $RH_{i,lam}$ approximately 30 % below the $RH_{i,lam}$
of the onset reported by Welti et al. (2009). The deviation may partly result from uncertainty related to the lamina conditions
as discussed in the following section (cf. §4.3). Welti et al. (2009) reports a temperature dependence of the freezing onset
for -40 °C < $T_{lam}$ < -30 °C, where one might expect $T_{lam}$ to greatly influence the results. DeMott et al. (2018) also reports a
strong sensitivity of NX-illite to $RH_{liq,lam}$ and suggests that it might be partly responsible for the wide range of AF obtained
for NX-illite experiments conducted with CFDCs. In addition, Hiranuma et al. (2015) mention that using different sample
preparation techniques and measurement methods can result in different AF even when identical test samples are used. In our
case, although we used the same sample as Welti et al. (2009) (Arginotec, NX Nanopowder), we followed two different sample
preparation methods which could also party explain the deviation observed. Indeed, Welti et al. (2009) used a Fluidized Bed
Aerosol Generator (TSI model 3400A) to generate the particles, while in our case the particles were dry generated using dried
and filtered compressed air flowing vertically through a glass filter containing the dust sample, and were then mixed into a large
volume to homogenize the sample before being transported to the DMA. It is important to note that evaluating experimental
observations using other experimental observations comes with multiplicative uncertainty, and large deviations may result from
accumulated uncertainty for both instruments. Such observations further motivate thorough identification and quantification of
instrumental uncertainties which we discuss in the following section.

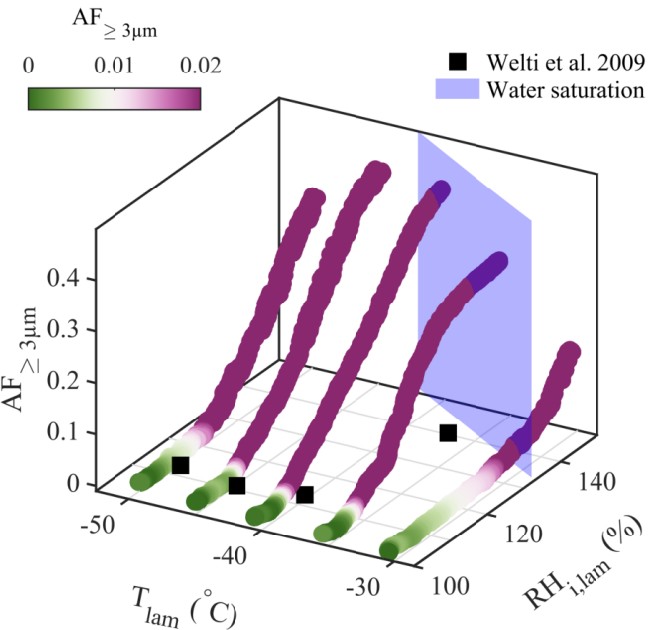

**Figure 9.** Heterogeneous freezing of size-selected 200 nm NX-illite particles. The AF of aerosols with $d_p \geq 3\ \mu m$ is plotted as a function of the average $T_{lam}$ and $RH_{i,lam}$. For comparison, the AF=1 % onset of similar particles reported in Welti et al. (2009) is shown as black triangles.

## 4 Measurement uncertainties

Chambers like PINCii come with inherent uncertainties that range from engineering choices to user determined experimental implementations and settings. For vertically oriented CFDCs, the condensed ice layer is one known source of uncertainty. Ice crystals can detach from the wall and fall through the chamber and into the detector, being mistaken for nucleated crystals – this is generally referred to as the chamber "background". In order to correct for this effect, checks are done during every experiment by sampling from clean, particle-free air. The measured background concentration is then subtracted from the concentration measured during ambient/sample measurements. The intermittency and averaging method used for both background and sample measurements are user-determined and should be optimized for the systems being studied. Another source of uncertainty related to vertically oriented CFDCs is the deviation of the lamina position. Indeed, the temperature gradient applied to create the supersaturation condition can slightly shift the lamina position, which is then no longer at the center of the chamber (Fig. 2). In addition, steady state modeling shows that, in certain cases where the temperature gradient is large, buoyancy processes become important and can lead to flow reversal (Rogers, 1988; Garimella et al., 2016), although as shown in Fig. 3, this also depends on the thickness of the ice layer. Given these inherent sources of uncertainty, it is important to quantify the fraction of particles exposed to lamina conditions that deviate from the prediction, as well as the fraction of particles lost within the chamber. In addition, the complex combination of PID-controlled heating pads and pulse-injected refrigerant may result



in wall temperature fluctuations which also increase uncertainty in the temperature and humidity conditions to which particles are exposed.

## 4.1 Background

The ice-coating on the chamber walls is one of the main limitations of vertical CFDC instruments. The quality and durability of the ice coatings determine the INP detection limits and the duration of instrument operation. Because the temperature gradient
results in vapor diffusion from the warm wall to the cold wall, and thus in a net evaporation of the warm wall's ice layer, no CFDC has an unbounded running time. Typical operational times of four to six hours between re-icing can be expected for CFDCs (Chou, 2011; Garimella et al., 2016; Lacher et al., 2017; Brunner and Kanji, 2021). However, these operational times are greatly affected by the initial quality of the ice layer. When the ice layers become unstable and increase the background, the CFDCs need to be warmed, drained, re-cooled and re-iced to continue experiments.

Here we shortly present the specifics of the icing procedure used with PINCii so that it may be adopted for future use. First, the main chamber walls are fixed at -23 °C, while the evaporation section is fixed to -20 °C. Water is then flowed into the chamber via the exit hole (k in Fig. 1), until the water level sensor detects water and stops the filling. After a delay of 5 seconds, the water is drained from the chamber using the same exit hole. When water is flowed into the chamber, it begins to freeze on the walls and the latent heat release warms the walls to nearly 0 °C. To avoid a too rapid cooling, the setpoint temperatures
are changed from -23 and -20 °C to -5 °C, and then decreased by 5 °C increments where the temperature is stabilized at each interval until the desired experimental conditions are reached. Throughout the process, a 10 L·min$^{-1}$ flow of dry nitrogen is introduced through the sheath flow inlets in order to sublimate structural heterogeneities from the ice coating. Note that, in the field, the dry nitrogen is sometimes replaced by dry, particle-free air. When the desired experimental conditions are achieved, the outlet (which tends to accumulate water) is dried before the OPC is re-attached. Although no direct measurements were
realized, results from background measurements show that this icing procedure tends to stabilize the ice layers and lead to low and stable backgrounds.

Fig. 10 shows the evolution of the background counts from exceptionally good to mediocre after successive RH$_{i,lam}$ ramps at different temperatures. The shown data originates from two identical experiments that were conducted during a measurement campaign (RH$_{i,lam}$ ramped from 110 to 160 % at T$_{lam}$ -26, -38 and -50 °C, each experiment took place over the course of ca.
3.5 h). Fig. 10 shows the background counts with a new ice layer before the first ramp and at the end of the experiments after three ramps, when the ice layer was deteriorated. With a fresh ice layer the background of PINCii is exceptionally low, with 67 % of all data points (standard OPC sampling rate of 0.2 Hz) being zero counts (0 #·L$^{-1}$) and 90 % percent of all counts being $\leq 4.8$ #·L$^{-1}$. However, after several RH$_{i,lam}$ ramps, the ice layer clearly deteriorated. The authors want to emphasize that the chamber conditions during these experiments are far from ideal for preserving a mint layer of ice, but they do represent
a real-world example and potential users should be aware that the ice layer conditions evolve when conducting experiments. Overall, if the ice layer in PINCii is in good condition, the background is very stable and at the lower end of background measurements reported from earlier CFDCs, which typically range from 1 to 10 counts·L$^{-1}$ (Chou, 2011), and is within the same order of magnitude as backgrounds obtained in more recent studies (Boose et al., 2016; Lacher et al., 2021).





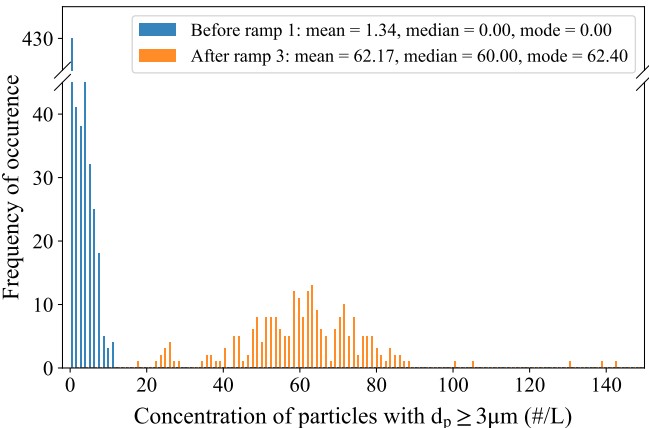

**Figure 10.** Histogram of background counts before and after successive $RH_{i,lam}$ ramps from 110 to 160 % at $T_{lam}$ -26, -38 and -50 °C. Note that at the end of ramp 3, PINCii had been running for approximately 3.5 h. The statistical values denoted in the figure legend have the same unit as the x-axis.

### 4.2 Particle loss

Particle losses may arise depending on how the sample of interest travels to and through the chamber, and if there are interactions with surfaces, other particles, and/or buoyancy or turbulence within the chamber. To quantify particle losses, experiments were conducted by injecting particles of known concentration and measuring the difference at the exit of the chamber. For these experiments, the total particle concentration was measured using a CPC at the chamber inlet ($CPC_{in}$, TSI model 3775) and the output concentration was measured at the chamber exit ($CPC_{out}$, TSI model 3787), where the OPC is usually mounted.

The ratio PL represents the fraction of particles that are lost and is defined as:

$$PL = \frac{CPC_{in} - (CPC_{out} \times df)}{CPC_{in}} \tag{1}$$

where $df$ is the dilution factor corresponding to the ratio between the total flow rate exiting PINCii and the sample flow rate entering the chamber ($df$=10). The CPC concentrations were adjusted at the beginning of each experiment to take into account any offset between the CPCs. Particle loss experiments were performed over the accessible range of thermodynamic conditions

within PINCii using 100 nm PolyStyrene Latex (PSL) spheres. PSL spheres were chosen because they are largely hydrophobic and non-ice active, and thus minimize nucleation that could result in scavenging.

The results presented in Fig. 11 show that negligible particle loss is observed for most chamber conditions, except when, (i) the saturation condition exceeds droplet breakthrough, and (ii) $T_{lam}$ < -37 °C and $RH_{liq,lam}$ > 100 %, where PL up to 40 % is observed. It is important to mention that meaningful ice nucleation experiments cannot be conducted in this regime. At

warmer temperatures, beyond droplet breakthrough, a phase discriminating detector could be used. On the other hand, the colder temperatures are within the realm of homogeneous freezing, and thus the link between INP and observed ice above water saturation is ill-defined. It is also notable that particle scavenging from nucleated particles is not observed below the



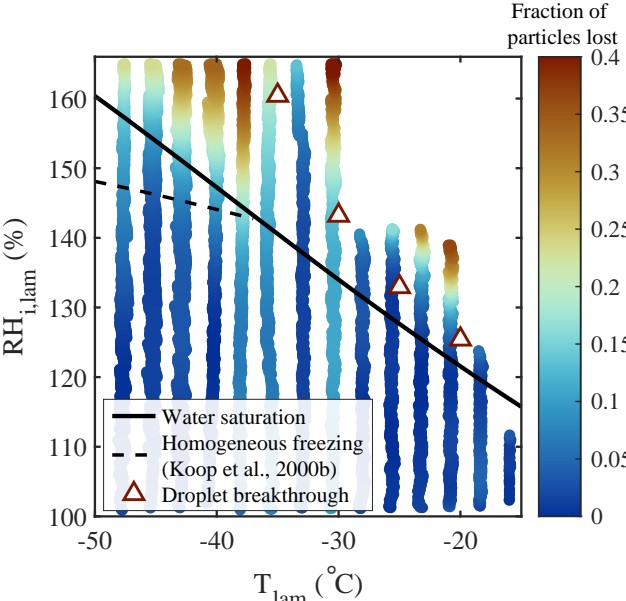

**Figure 11.** Fraction of particles lost (PL, color-bar) measured using 100 nm PSL spheres. For the experiments, RH was increased at 1 % min$^{-1}$, beginning from RH$_{i,lam}$=100 %, for fixed T$_{lam}$. Water saturation (solid black line), droplet breakthrough (triangles) and homogeneous freezing (dashed black line, calculated with $\Delta a_w = 0.2946$ following Koop et al. (2000b)) curves are added for supplementary information.

droplet breakthrough curve (at 100 % < RH$_{liq,lam}$ < RH$_{DB}$). This is strong evidence that the laminar sample flow minimizes particle-particle interactions.

Additional experiments similar to those described in DeMott et al. (2015) were performed to evaluate whether particles deviate from the laminar flow. Pulses of 240 nm PSL spheres were input into the chamber by turning on and off a DMA, and the measured CPC$_{in}$ and CPC$_{out}$ concentrations are plotted in Fig. 12. For the duration of the experiment, the main chamber was run at fixed lamina conditions (T$_{lam}$ = -50 °C & RH$_{i,lam}$ = 160 %) with the evaporation section isothermally fixed at T = -39.6 °C and RH$_i$ = 100 %.

When CPC$_{out}$ is scaled for the sheath flow dilution (by multiplying by 10) and shifted in time by 15 s (the chamber residence time at these conditions), the result clearly demonstrates the coherence of the input pulse, even upon exiting (Fig. 12a). However, a more detailed view of the first pulse shows slight variations in the peak shape, including an elongated tail resulting from delayed particles that have passed through the chamber more slowly than expected if they were travelling only within the sample lamina (Fig. 12b). Quantification of the fraction of delayed particles which most likely do not experience lamina condition is done by assessing the difference in the pulse distributions. Although a systematic calculation would require complex convolutions of the particle and velocity distributions, the task is simplified thanks to the experimental time resolution. As illustrated in Fig. 2, particles in the lamina do not travel at uniform velocity, nor do they always have the maximum flow velocity within the chamber. However, given the chamber residence time, the potential differences within the lamina and between the lamina

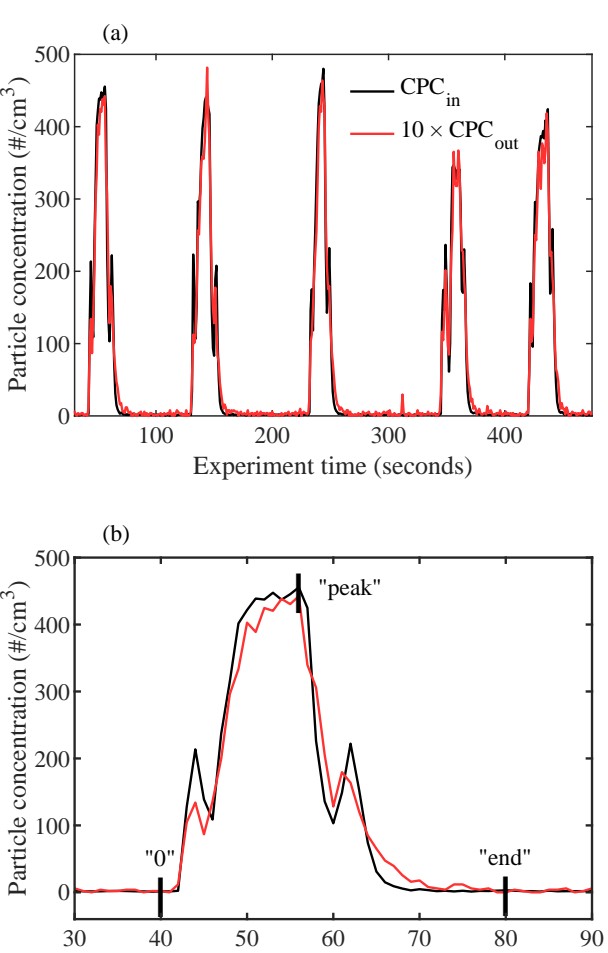

**Figure 12.** Time series of measured particle concentrations at the entrance $CPC_{in}$ (black) and exit $CPC_{out}$ (red) of PINCii for experiments using inputs of 240 nm PSL spheres. The main chamber was run at fixed lamina conditions ($T_{lam}$ = -50 °C & $RH_{i,lam}$ = 160 %) with the evaporation section isothermally fixed at T = -39.6 °C and $RH_i$ = 100 %. The exiting pulses are time shifted by 15 s in order to overlap the pulse onset. In panel (b), the first pulse pair is highlighted, with the integration limits from equations 2, 3 and 4 illustrated as an example ("0", "*end*", "*peak*").

and the maximum flow velocity would result in a spread of arrival times of less than the 1 s CPC time resolution. This means

that, for measurement purposes, the lamina velocity can be considered constant. In addition, the input and output pulses should





be identical, and only diluted and shifted in time to achieve ideal lamina flow within the instrument resolution. Moreover, this allows us to utilize the onset of each $CPC_{out}$ pulse to represent particles that have travelled through the sample chamber within the sample lamina. Following Eq. 1, the total integrated particle loss can be calculated as,

$$PL_{tot} = \frac{\int_0^{end} CPC_{in} - \int_0^{end} CPC_{out}}{\int_0^{end} CPC_{in}}, \tag{2}$$

where the pulse is integrated from the onset (0) until concentrations return to pre-pulse values ($end$), as shown in Fig. 12b. A percentage delayed fraction DF can then be calculated,

$$DF = \frac{\int_0^{peak} Delayed}{\int_0^{peak} CPC_{in}} \times 100, \tag{3}$$

where $\int_0^{peak} Delayed$ is calculated as,

$$\int_0^{peak} Delayed = (\int_0^{peak} CPC_{in} - \int_0^{peak} CPC_{out}) \times (1 - PL_{tot}). \tag{4}$$

When the delayed quantity DF is calculated for the five presented pulse pairs, an average of 9.1 % of particles are found to be delayed, with a minimum of 8.0 % and a maximum of 11.1 %, which is two times lower than the ratio of delayed particles reported by Garimella et al. (2017) when they conducted similar pulse experiments with ZINC and SPIN. While these numbers are only valid at the experimental conditions used here, they represent the most turbulent achievable conditions obtained with typical flow rates. Furthermore, following the assumption that for each pulse, the particle onset represents particles that travelled within the lamina, it is then likely that delayed particles were dispersed towards the regions of lower velocity, which are also the regions with lower $RH_i$ than the lamina (Fig. 2). The delayed particles are therefore less likely to initiate nucleation, especially in deposition mode. In other words, the small percentage of delayed particles that travel outside the lamina are unlikely to affect experimental results by showing a pre-activation, but may lead to an underestimation of the INP concentrations proportional to the fraction of particles delayed (Fig. 11).

## 4.3 Temperature control

The chamber wall temperatures control both $T_{lam}$ and $RH_{i,lam}$. Variations and uncertainties in the wall temperatures propagate through the chamber(s) and can result in variability in the sample conditions, affecting parameters such as particle activation and calculation of uncertainties. Diagnosing unexpected (in)activation of particles and spurious INP counts relies on reliable temperature stability and well-resolved wall temperature monitoring. To achieve this, current PINCii models include a high spatial distribution of 58 Type-K thermocouples that monitor temperatures at 1Hz (Fig. 1). This leads to a high spatial resolution of the temperature distribution inside the chamber and allows us to quantitatively assess the range of deviation. The high spatial resolution temperature monitoring can also be used to carefully assess forcing extremes (cf., §3.3). In Fig. 13, the standard errors of $T_{lam}$ and $RH_{i,lam}$ calculated from wall temperature profiles measured over the full operational range of PINCii are presented. The temperature measurements were made during the previously presented droplet nucleation (Fig. 5),



homogeneous freezing (Figs. 8 & A1) and particle loss (Fig. 11) experiments. Standard errors are inserted into an empty $RH_{i,lam}$, $T_{lam}$ coordinate matrix, and missing points are calculated using a nearest neighbors linear interpolation. Results show that uncertainties in $RH_{i,lam}$ are most important at $T_{lam} < -40$ °C and $RH_{i,lam} > 140$ % where the standard error can exceed 2 %. On the other hand, uncertainties in $T_{lam}$ are most apparent at warm temperatures ($T_{lam} > -25$ °C), where the standard error can exceed 0.35 °C. The lamina uncertainties are primarily the result of temperature inhomogeneities on the chamber

walls caused by strong cooling near the coolant injection points. PINCii was designed with multiple injection points in order to minimize these effects, but our results show that improvements are still needed. Nevertheless, our results also emphasize the advantages of having a high spatial temperature resolution to help identify and potentially correct temperature heterogeneity during experiments. Note that the uncertainties presented here only account for the spatial variations in temperature and RH and thus represent ideal (minimum) uncertainties. The actual measurement uncertainties are higher and we recommend quantifying

them on an experiment-by-experiment basis since they highly depend on the operating mode used and the actual temperature conditions.



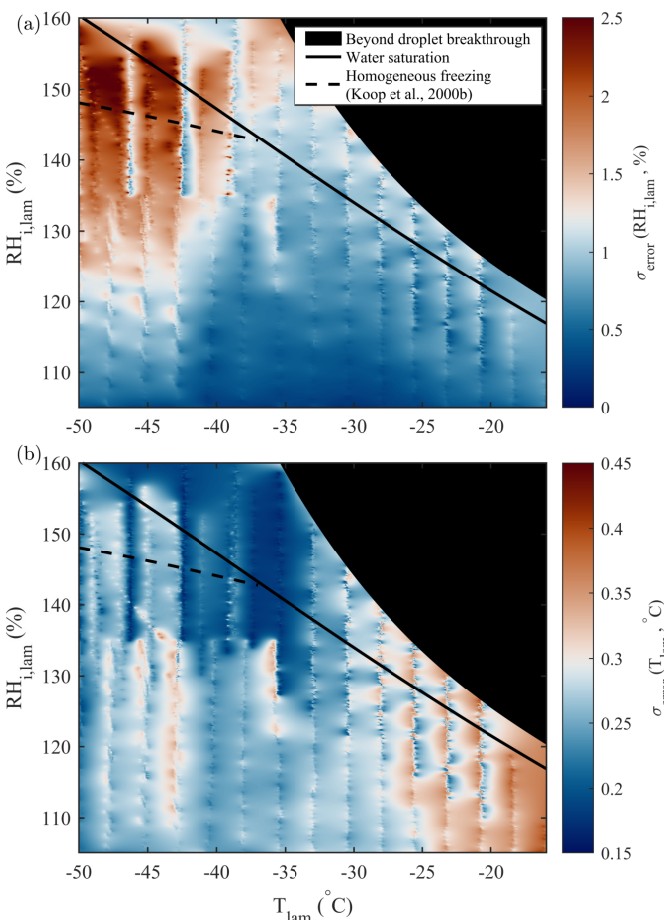

**Figure 13.** Standard error of (a) $RH_{i,lam}$ and (b) $T_{lam}$ calculated from the scanning done during droplet nucleation (Fig. 5), homogeneous freezing (Figs. 8 & A1) and particle loss (Fig. 11) experiments. The theoretical curves for water saturation (solid black line) and homogeneous freezing (dashed black line, calculated with $\Delta a_w = 0.2946$ following Koop et al. (2000b)) are added for supplementary information.

## 5 Conclusions

This study presents the design, working principles, and operating capabilities of the newly developed PINCii. Experimental results are used to assess the system operation and quantify system uncertainty. The upgraded capabilities of PINCii relative to previous generations of deployable CFDCs are highlighted.

Engineering upgrades to the design include an elongated main chamber that enables a longer aerosol residence time ($\sim$15 s) and therefore enhanced nucleation and growth. This may be helpful for experiments performed at cold temperatures (T $\lesssim -45°C$) where ice crystals may not grow to size thresholds commonly used for ice detection (Fig. 8; Welti et al., 2020). An





elongated evaporation section improves phase differentiation and thus results in more reliable data output at water-saturated

conditions.

In addition to its cascade cooling system, PINCii has a large array of wall mounted heating pads. Coupled with the evaporation section, whose temperature can be controlled independently, PINCii has the potential to be operated in a low temperature CCNC mode, wherein a temperature gradient can also be maintained along the evaporation section. This feature can also be used for ice nucleation experiments below water saturation (homogeneous freezing of solutions or deposition freezing), where

phase differentiation is not necessary.

PINCii has a dense thermocouple network embedded within all chamber walls (Fig. 1) that offers reliable monitoring of wall temperatures and is used to analyze data from homogeneous freezing experiments in new ways. We find that measured homogeneous freezing onsets agree better with the theoretical onset from Koop et al. (2000b) when using the most extreme lamina conditions present within the chamber (greatest $RH_{i,lam}$), representing conditions most likely to trigger nucleation

(Figs. 8c & A1).

The Rogers (1988) thermodynamic model has been updated to account for the ice layers coating the chamber walls. The update shifts the predicted $T_{lam}$, $RH_{lam}$ and flow profile across the chamber, and deviations up to 1 °C in $T_{lam}$ are found when compared to the model output absent condensed ice layers. In addition, the updated model predicts a laminar flow profile with negligible flow reversal when accounting for the ice layers.

Results from a sequence of experiments show good agreement with applicable theories or previous studies and are summarized in Fig. 14. In this figure, the activation onset obtained for each experiments is represented. For the homogeneous freezing experiments (200 nm NaCl and 200 nm natural salt sample from Qaidam basin, China in Fig. 14), the ice nucleation onset was defined as the median of the inflection points obtained for each activation curve as done in Fig 8 (see A1 for more information). For the heterogeneous freezing experiments with 200 nm NX-illite particles, the ice nucleation onset was defined as AF=1 %

to compare our results to Welti et al. (2009). When the AF could not be calculated because the experiments did not include running a CPC in parallel to PINCii, the activation onset was defined as a fixed value. For the droplet formation experiments conducted with ambient polydisperse particles, the onset was defined as the ratio of activated droplets equal to 0.1 (see Droplet nucleation in Fig. 14). For the droplet breakthrough experiments, the onset was defined as 1 droplet·cm$^{-3}$. Fig. 14 also shows the results of the deliquescence experiments conducted with 200 nm NaCl particles at both -35 and -36 °C. Overall, results

show that droplet formation, deliquescence, homogeneous and heterogeneous freezing onsets were observed at the predicted conditions (Köhler, 1936; Koop et al., 2000a, b; Welti et al., 2009).





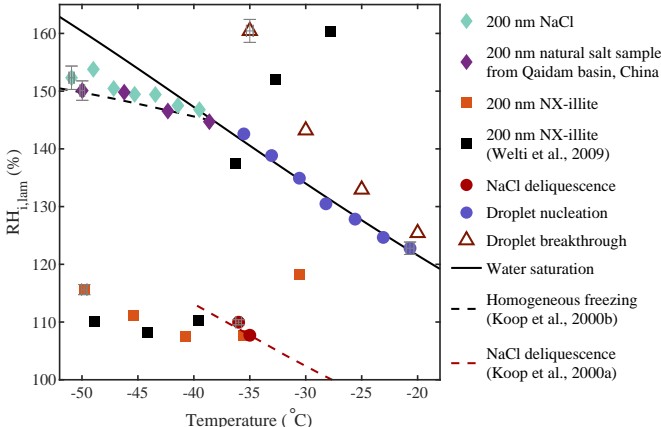

**Figure 14.** Overview of the ice nucleation onsets measured during homogeneous freezing experiments (diamonds; also in Figs. 8c and A1) and heterogeneous freezing experiments (squares; also in Fig. 9), as well as results from NaCl deliquescence experiments (red circles; also in Fig. 6), droplet nucleation experiments (blue circles; also in Fig. 5) and droplet breakthrough experiments (red triangles; also in Fig. 7). The theoretical curves for water saturation (solid black line), homogeneous freezing (dashed black line, calculated with $\Delta a_w = 0.2946$ following Koop et al. (2000b)) and NaCl deliquescence (red dashed line, digitized from Koop et al. (2000a)) were added for supplementary information. Error bars representing the standard errors in $T_{lam}$ and $RH_{i,lam}$ were added on a selection of data points covering each experiment.

PINCii is a flexible instrument able to reach a wide range of thermodynamic conditions (Fig. 4) while remaining transportable and capable of conducting nucleation experiments in both ice and liquid phase. The analysis of the main instrumental uncertainties highlights that PINCii can operate with very low background signal and minimal particle losses (Figs. 10 & 11).

This means that PINCii is suited for sampling from sources with low INP concentrations.



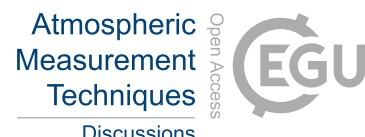

## Appendix A

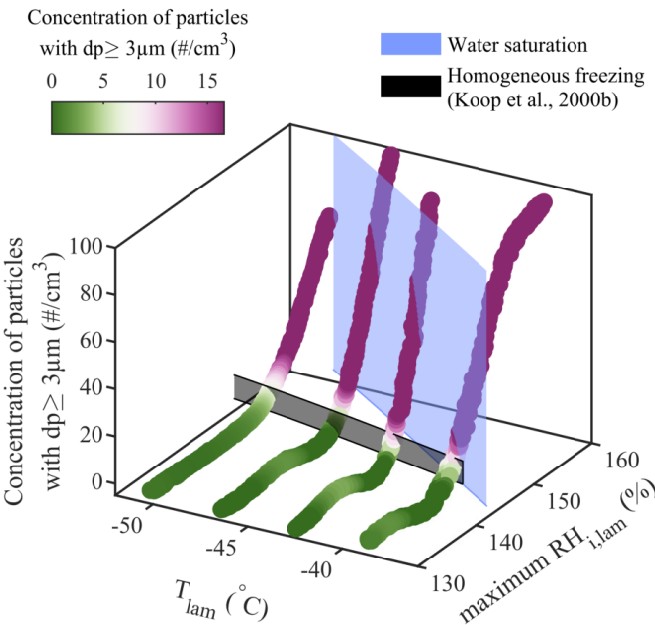

**Figure A1.** Homogeneous freezing of 200 nm particles generated from a natural salt sample collected in the Qaidam basin, China. Note that, due to an instrumental malfunction, the CPC was not running and thus the AF could not be calculated for this experiment. The concentration of aerosols with $d_p \geq 3$ µm is plotted as a function of the average $T_{lam}$ and the maximum $RH_{i,lam}$, and the color scale is used to represent changes in the concentration. The white region in the color bars represents the ice nucleation onset which was estimated using the median of the inflection points obtained for each activation curve (see A1 for more information). The theoretical curves for water saturation (blue shading) and homogeneous freezing (black shading, $\Delta a_w = 0.2946$, Koop et al., 2000b) are projected into the 3-d space as in Fig. 8.

## A1 Estimation of the ice nucleation onsets

Here we describe the method followed to estimate the ice nucleation onsets represented as the white region in the color bars of Fig.8. First the activation curves were resampled and averaged using a 10-point moving average (circles in Fig.A2a). Then a
Gaussian filter was used on the resampled and averaged data to obtain a continuous, smooth curve (solid blue line in Fig.A2a) from which the second derivative was calculated (orange dashed line in Fig.A2a). The inflection point was then defined as the maximum of the second derivative (vertical dashed line in Fig.A2a). This procedure was applied to each activation curve of the experiment as shown in Fig.A2b, and finally the median of the inflection points was calculated and used as an estimate of the ice nucleation onset in Fig.8.





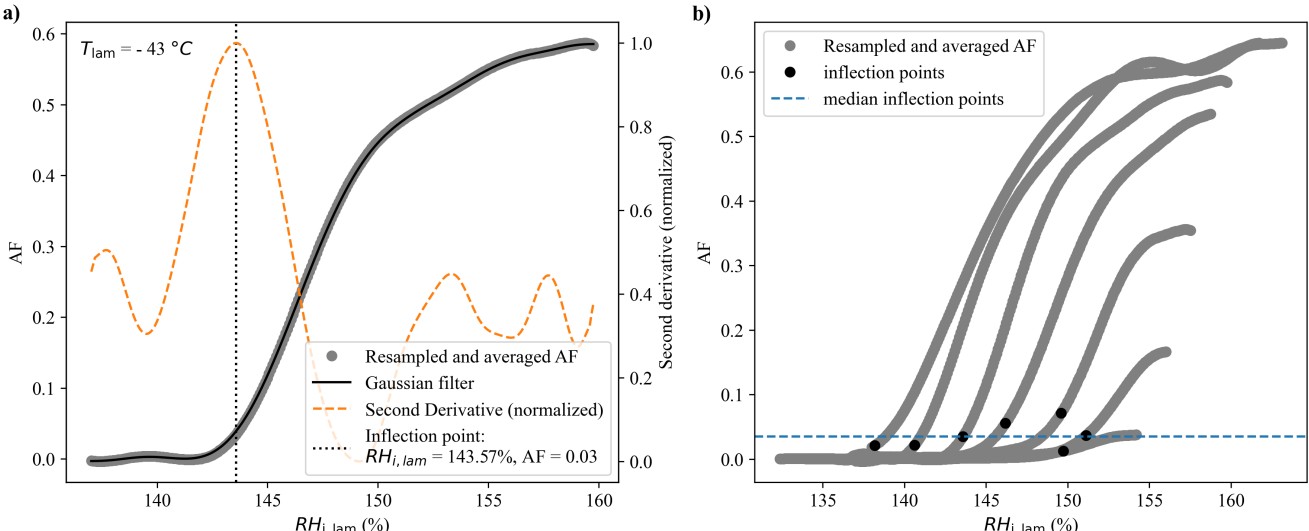

**Figure A2.** a) Example of activation curve obtained during the RH ramp conducted at -43 °C, where the colored circles represent the data after it was resampled and averaged using a 10-point moving average. The solid blue line represents the Gaussian filter applied to the resampled data and the orange dashed line represents the normalized second derivative calculated from the Gaussian filter. The inflection point, highlighted with the vertical dashed line, was defined as the maximum of the normalized second derivative. b) All activation curves obtained during the experiments, with the black circles representing the inflection points obtained for each curve and the blue dashed line representing the median of these inflection points.

## 475 A2 Ice crystal growth

Ice crystal growth is determined using the approach presented in Welti et al. (2020), which is based on the mass growth rate of ice crystals given in Rogers and Yau (1989) and the simplifying assumption of spherical ice crystals:

$$r = \sqrt{r_0^2 + 2 \cdot \alpha \cdot \frac{S_i - 1}{\rho_i \cdot (F_k + F_d)} \cdot t_{res}}, \tag{A1}$$

with the time-dependent ice crystal radius $r$, the seed particle radius $r_0$, the mass accommodation coefficient $\alpha$, the saturation
ratio with respect to ice $S_i$, the mass density of ice $\rho_i$, and the residence time in the chamber $t_{res}$. $F_k = (\frac{L_s}{R_v T} - 1) \cdot \frac{L_s}{KT}$ with the latent heat of sublimation $L_s$, the individual gas constant for water vapour $R_v$, and the the thermal conductivity of moist air $K$. $F_d = \frac{R_v T}{D_v p_{sat,i}}$ with the diffusivity of water vapor in air $D_v$, and the saturation vapour pressure over ice $p_{sat,i}$. The values of the aforementioned variables were determined according to the respective references in Welti et al. (2020), except for $R_v$, which was determined from Skrotzki et al. (2013). It should be noted that the mass accommodation coefficient $\alpha$ represents
the largest uncertainty in calculating the ice crystal diameter. Skrotzki et al. (2013) report a range of $0.2 - 1$ for $\alpha$ at T = 190 - 235 K. We chose $\alpha = 0.3$ in order to make the results comparable to Burkert-Kohn et al. (2017), who present ice crystal growth calculations for PINCii's predecessor PINC and use $\alpha = 0.3$.

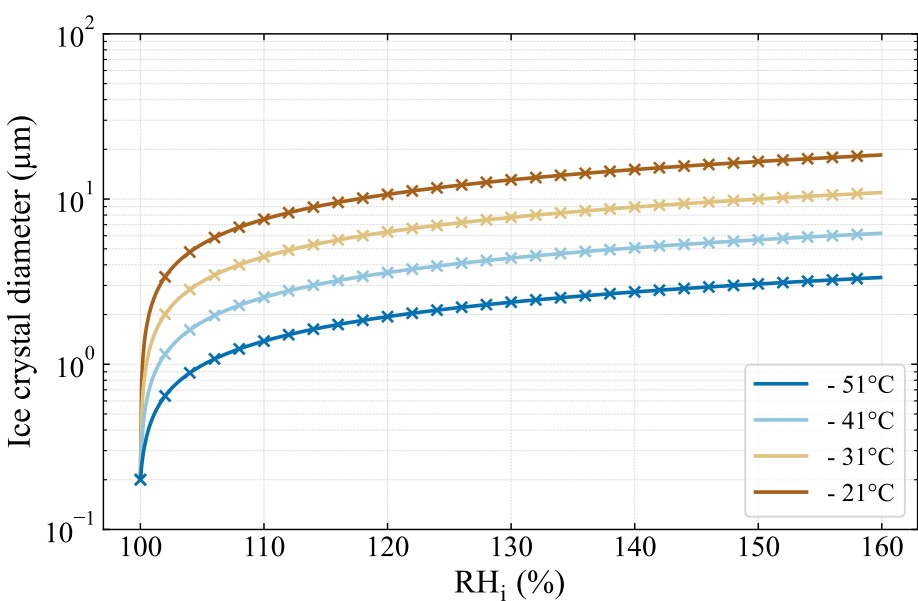

**Figure A3.** Ice crystal growth calculations for the typical residence time of 15 s in PINCii following Welti et al. (2020), and using a mass accommodation coefficient of 0.3 and initial seed particle diameter of 200 nm. Line markers (x) are spaced at 2% $RH_i$ increments.



*Data availability.* The data shown in the paper are available upon request from the corresponding authors.

*Author contributions.* The authorship reflects the principal parties that established a PINCii technology sharing agreement, a project that
began nearly a decade ago as a collaboration between ETH-Zurich, TROPOS, the University of Gothenburg, Lund University, Aarhus
University, and the University of Helsinki. All parties contributed resources and technical expertise to the PINCii development. The original
PINCii design and engineering upgrades were suggested at a workshop hosted at ETH-Zurich, with O.S., leading the workshop and presenting
his original ZINC and PINC designs, which established the basis for PINCii. All authors contributed to discussions and suggestions for
engineering upgrades. Q.T.N., and M.B., helped engineer and requisition the first chamber components. E.S.T., D.C., J.D., Z.B., and Y.U.,
assembled the first PINCii chambers. D.C., Y.W., Z.B., and E.S.T., wrote the PINCii control and data management programs. D.C., Z.B., Y.W.,
J.D., and E.S.T., first ran PINCii experiments and established the operating procedure. D.C. ran and collected the data from all experiments
presented herein. D.C., Z.B., Y.W., Z.A.K., M.H., J.D., and E.S.T., reviewed and discussed all the results from the experiments. D.C., Z.B.,
M.H., and E.S.T, wrote the manuscript. All authors read, reviewed and commented the manuscript.

*Competing interests.* The authors declare no competing interests.

*Acknowledgements.* The PINCii project has been supported by the Nordic Centre of Excellence CRAICC (Cryosphere-Atmosphere Interac-
tions in a Changing Arctic Climate). EST and DC have been supported by the Swedish Research Councils, VR (2013-05153, 2020-03497)
and FORMAS (2017-00564). EST, DC and BSv have been supported by the Swedish Strategic Research Area MERGE. EST and ZAK
thank the Gothenburg Air and Climate Network for funding to support ZAK's sabbatical time during which ZAK participated in many of
the characterization experiments. Xiangrui Kong is acknowledged for collection and use of the Qaidam Basin natural salt samples. ZB, YW,
LA, MK, TP and JD have been supported by the Jane and Aatos Erkko Foundation, by the European Commission via "Climate Relevant
interactions and feedbacks: the key role of sea ice and Snow in the polar and global climate system (CRiceS, 101003826), by the Academy
of Finland via a Flagship programme for Atmospheric and Climate Competence Center (ACCC, 337549) and projects (334792, 340791,
333397, 329274, 328616, 352415, 345510), and by the Academy professorship funded by the Academy of Finland (302958).



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
