# Peer review of "Development and characterization of the Portable Ice Nucleation Chamber 2 (PINCii)"

_Atmospheric Measurement Techniques, 2023_

## Referee Comment (RC1)

**Referee comment: Development and characterization of the Portable Ice Nucleation Chamber 2 (PINCii), Castarède et al., 2023**

The manuscript presents a new instrument to measure ice-nucleating particle concentrations, both in the laboratory and the field. Even though there is already a high number of instruments of the same type, PINCii is a further improvement of the measurement technique. Furthermore, the authors discuss a new approach to analyze the data of a CFDC. To validate the instrument, multiple experiments including deliquescence, homogeneous freezing and heterogeneous freezing were performed and the results are compared to literature values from different studies.

Overall, the manuscript is well written and structured and gives detailed explanations of the work that has been done. However, there are some points, listed below, that would further improve the quality of the paper. My comments are structured in first more general comments and second minor comments.

**General comments**

- Given the length of the text, there is a quite high number of figures, such as e.g. in section 2.4. I would recommend to reduce the number of figures and instead explain the outcome of the measurements in more detail in the text. The figures can then be either deleted or moved to the appendix.

- Figure 8, 9 and A1 are very complex and hard to understand. They should be simplified to a 2D plot, with AF as a color scale. If the change in AF is the important parameter for defining the onset, then the color code can also be presented as Δ AF.

- How much does a potential underestimation of the number of ice crystals due to the binning of the OPC and the set ice threshold contribute to the total uncertainty of the INP concentration? The authors provide a calculation of the ice crystal growth, however, this assumes a spherical ice crystal and a constant mass accommodation coefficient of 0.3. Based on this calculation, the threshold for ice crystal detection in the OPC was set. However, since the binning of the OPC is quite broad, some particles might not be counted as ice (or droplets), because they are not detected in the respective channel of the OPC. This could be especially relevant when measuring INPs in low concentrations.

**Minor comments**

- L6: The phrase "very low concentrations" should be supported with numbers of the range of the limit of detection
- L21-22: In line 21 you are writing "heterogeneous nucleation", however, in the following sentence in L22 you call it "heterogeneous ice nucleation". You should stick to one term, preferably the second one.
- L39: Mention that the CFDC-IAS has a cylindrical shape
- Figure 1: A list inside the figure explaining the letters (a) to (k) would help for an easier understanding. The color of (f) (refrigerant cooling coil pipes) should be changed, because it is difficult to differentiate it from the other items.
- L89: Briefly explain what ETH-IODE is and what it is used for
- L101: Explain the abbreviation R23
- L117: How much longer is the main chamber of PINCii compared to other CFDCs? You should give at least a range of numbers

- Figure 5: The symbols of the first and second experiment are very hard to distinguish in the plot. As it is written in the text, the data are presented as normalized values, so it might not be needed to present them in different symbols. If the authors think, that it gives the reader some value to know which data points were recorded on which day, they should divide the figure in two sub plots. It might be also beneficial to mark the range in which the data points represent either an activated cloud droplet or an ice crystal.
- L234-239: The authors should elaborate a bit more the outcome of figure 8(a) by giving numbers e.g. at which RH_lam activation happens for different temperatures and how much it differs from the Koop line.
- L247-248: How is it seen that some ice crystals did not grow to 5µm. I guess it can be seen by the ice threshold that is shifting to a lower RH_lam from Fig. 8(a) to 8(b). However, a short note on that might be helpful for the reader.
- L274-276: Replace one of the "significantly"
- Figure 9: In the caption, replace "triangles" by "squares"
- L329: Add a short note why the rapid cooling should be avoided
- L237: "exceptionally" and "mediocre" need to be defined in terms of values
- L341: Remove "exceptionally"
- L343: A short discussion about the background concentration after 3 ramps is missing
- Figure 11: Was the droplet break through only measured for four of the scans?
- L465: Rephrase to "sampling from sources with INP concentrations as low as …"
- Figure A1: This figure is mentioned quite often in the text. Therefore, I suggest to move it from the appendix to the main text.

---

## Author Comment (AC1)

The paper by Castarède et al. (2023) is a nice addition of ice nucleation chamber to the community. Our recent chamber (Kulkarni et al. 2020; see below the reference) also employs long evaporation (nucleation) section ( 0.45m), and we found that this feature evaporates supercooled droplets very efficiently.  Figure 5 in Castarède et al. (2023) paper indicates droplet formation at RHw= 100%. These observations are also reflected in the droplet breakthrough section (3.2) and Figure 7. Just wondering and curious, as the evaporation section is long (0.43m) in PINCii, does the evaporation section evaporate the droplets? What are the evaporation section conditions (T and RH) while performing measurements shown in Figures 5 and 7? In our study, we maintain the evaporation section at RHice = 100% and isothermal experimental temperature conditions.

Reference:

Kulkarni, G., Hiranuma, N., Möhler, O., Höhler, K., China, S., Cziczo, D. J., and DeMott, P. J.: A new method for operating a continuous-flow diffusion chamber to investigate immersion freezing: assessment and performance study, Atmos. Meas. Tech., 13, 6631–6643, https://doi.org/10.5194/amt-13-6631-2020, 2020.

**Response**: Thank you for your comments and questions.

For the droplet activation experiments shown in Figure 5, the evaporation section was not running in the normal "evaporating mode", but was running with the same temperature gradient as in the main chamber. This was done in order to keep the temperature gradient along the entire chamber length (1.43 m) to allow droplet activation and growth, which we show in Figure 5. It seems like this was not clear in section 3.1, also from the referees' comments, so we added more information to the text:

L167 "In this work, we use this feature to show that the chamber can actively grow droplets."

And we added a sentence at L169:

"First, we study the activation of polydisperse ambient aerosol particles, and then we present a deliquescence experiment with 200 nm Sodium Chloride (NaCl) particles. For both experiment types, the temperature gradient along the main chamber is extended to the evaporation section, so the evaporation section is no longer evaporating droplets."

Note that it is different for the droplet breakthrough experiments presented in Figure 7, where the evaporation section is running in the normal "evaporating mode" described L73-75. As in your study, the evaporation section is held at isothermal temperature conditions (at the same temperature as the warm wall) and RHi = 100% to evaporate the droplets, at least below droplet breakthrough.

---

## Author Comment (AC2)

**General Comments**

This is a fine paper about very good technical improvements of an ice nucleation device. It is excellent to have the new device documented in such detail, making it both useful for a variety of studies and potentially reproducible. I will say upfront (the authors may disagree) that I find some of the purported unique features of the instrument to be overplayed in places, in consideration of what is already in the literature for CFDCs that are fully as portable as this one and have even been operated in extended design and with low flow rates to approach lower operational temperatures. It is sometimes difficult to understand what is new for PINC versus what is new for all operational CFDCs. Otherwise, the paper is very well-written and I have mostly a number of minor questions, comments and editorial suggestions listed below. My recommendation is that this paper could be modestly revised and be fully acceptable for publication.

**Response**: We thank the referee for the constructive comments and feedback that will improve the manuscript. We include our responses below, in the context of individual comments.

**Specific Comments**

**Abstract**

I did not understand the "new method to analyze CFDC data." Can you clarify? is it meant that the evaporation section can have a temperature change or not and thus permit monitoring droplet activation and allow additional ice growth at low temperatures? I do not see this as a new development based on references given in the paper and below.

**Response**: Here we refer to the new method used to calculate ice nucleating conditions (and thus ice nucleation onset) based on the most extreme lamina conditions (greatest $RH_{i,lam}$) that is used to produce Figures 8a and A1. We modified the sentence in the abstract to avoid confusion:

"This feature is used to introduce and discuss a new method for analyzing CFDC data based on the most extreme lamina conditions present within the chamber, which represent conditions most likely to trigger ice nucleation."

**Introduction**

Lines 51-52: I never imagine uncertainties being low enough in a CFDC to serve as a CCNc, and this paper does not give me confidence that it is so (see specific comment later). Hence, I question even making this assertion without proof.

**Response**: We propose rephrasing the sentence:

"In addition, we show that the entire chamber can be run with a temperature gradient (including the evaporation section), meaning that the chamber can be used to extend ice crystal growth or to study droplet activation processes at cold temperatures."

As well as the sentence lines 165-166:

"In addition to its use for ice nucleation experiments, PINCii can be used for droplet activation experiments whereby a temperature gradient is maintained along the entire chamber length, thus retaining droplets which can reach the optical detector [...]"

We propose removing the sentence lines 171-173 (starting with "Further characterisation to evaluate PINCii's used as a CCNC [...]"), and rephrasing lines 202-205:

"These results suggest that PINCii can be used to study droplet activation at cold temperatures. Such a feature could potentially be further developed to use PINCii as a low temperature cloud condensation nuclei counter (CCNC). This would however require more experiments with well characterized salts of known size distribution(s) and chemical composition(s), which we do not explore in this work. It would also require an optical detector with more size channels at the exit of the chamber, and a more rigorous characterization of the $RH_{lam}$ and its uncertainty."

And finally, lines 436-438:

"Coupled with the evaporation section, whose temperature can be controlled independently and thus wherein a temperature gradient can also be maintained, our results suggest that PINCii has the potential to be used to study droplet activation at low temperatures."

**Instrument design and operation**

Line 109: Please understand that I am not quibbling about the comprehensiveness of this list of instruments. Rather I want to note that aircraft CFDCs are also "portable", as are most IN instruments. The CSU CFDC has been flown in two basic configurations. In DeMott et al., (2003), a cascade refrigeration system was used, which then also transported for use in a mountaintop laboratory (Richardson et al., 2007). In later years (e.g., Barry et al., 2021), single stage refrigeration compressors have been used for mixed phase conditions. These instruments are portable in nature, being used in different lab and field scenarios on the ground or on ships as well (e.g., Knopf et al., 2021). And not to cross fine hairs, but the long version (Patnaude et al., 2021; Kasparoglu et al., 2022) has also previously been deployed to external laboratories for studies (e.g., DeMott et al. (2009). All such instruments are portable for surface-based deployments, even if one is transporting chillers and their fluid. This is in fact how the Handix commercial CFDC is constructed (Bi et al., 2019). I understand what PINCii stands for of course, but the portable distinction is a vague one and so "…operational portable CFDCs…" is really most CFDCs.

**Response**: We understand the referee's comment and we removed all mentions of "portable" (line 3 and line 109) unless it was used to explain PINCii's acronym.

Line 122: Here I will offer a minor quibble with the idea that the extended design of PINCii is a unique feature for deployable CFDCs, given my understanding of what can easily be deployed. In fact, the operation of such long-column instruments without using the evaporation section (i.e, continuing at the same wall temperatures) and at low flow rates has nearly fully been motivated by the known slow ice crystal "growth kinetics" at low temperatures (Patnaude et al., 2021; Kasparoglu et al., 2022), elucidated at length in the Richardson studies. It is a desirable feature of any CFDC deployed for low temperature studies, and only a few have featured this capability. "Compared to all but a few CFDCs…"?

**Response**: We agree with the referee's comment, and we propose to rephrase the sentence line 116:

"Compared to all but a few CFDCs, and in contrast to the first PINC, PINCii has an elongated design [...]"

Page 6, Section 2.4: Three minor things. What is most important for making this change in reverse flow results? Is it the narrower gap or the account for heat transfer through the ice, or what exactly? Since this is a desirable outcome, and not everyone achieves 1 mm of ice thickness in their current icing protocols, this is important to understand. Secondly, I suggest repeating here or in the figure captions that all calculation shown are for one total flow rate (10 lpm)? Finally, are the vapor pressure relations used for calculations stated anywhere?

**Response**: The change in the chamber width (due to the narrower gap) is the most important factor when looking at the changes in flow reversal. We propose rephrasing lines 147-157 to make it more clear. We also refer to the flow rate and the relation used for vapor pressure calculations, and we add a reference to Castarède (2021)'s thesis where the calculations performed for PINCii are detailed:

"Here we updated the model to account for the 1 mm thick ice layers on the side walls (hatched gray region, Fig. 2). This thickness value is calculated by measuring the total volume of water exiting the chamber after an icing / melting cycle and dividing it by the total surface of the chamber walls. Accounting for the 1 mm thick ice layers changes both the chamber width and the boundary conditions (i.e., the temperature of the ice layers). First, the machined 1 cm gap between the chamber walls becomes 0.8 cm and the total chamber volume is decreased by $\approx 20$ %. Then, the ice layers' temperatures are calculated accounting for the heat transfer between the aluminum chamber walls (in which the thermocouples are embedded) and the ice/gas interface, after which the chamber flow and thermodynamic conditions are calculated across these two ice layers. The detailed calculations performed for PINCii can be found in section 3.3.2 of the Castarède (2021) thesis, which is available digitally. The results are presented in Fig. 2, where the thermodynamic model output was calculated for a total flow rate of 10 lpm, and where the Clausius-Clapeyron relation was used to express the dependence of saturation vapor pressure on temperature. It is noteworthy that accounting for the ice layer thickness changes both the $T_{lam}$ and $RH_{lam}$, and simultaneously changes the lamina position and the velocity profile across the chamber (Figs.2 and 3). The reduction of the gap between the chamber walls due to the ice layers affects flow reversal (negative velocity), as seen in Fig. 3 where the velocity was represented as a function of the ice thickness increasing from 0 to 1 mm. While the standard model from Rogers (1988) predicts flow reversal due to buoyancy, as shown in Garimella et al. (2016), the updated model predicts a laminar flow profile with negligible reversal even when strong temperature gradients are applied between the chamber walls, as seen in Fig. 4 where the reversed flow fraction is presented for the full range of $T_{lam}$ and $RH_{lam}$ achievable with PINCii."

We also mention the flow rate in the figure's captions.

**Evaluating the chamber performance**

Line 168: An NaCl particle size of 200nm must represent a critical supersaturation that would be unresolvable from water saturation. Is that the point (yes based on sentence below, so can consolidate to say so right here)? What if you used 30 nm?

**Response**: The 200 nm NaCl particles were used for the deliquescence experiments, not the droplet activation experiments, which were conducted with polydisperse particles from ambient laboratory air. We rephrased line 168 to avoid misunderstanding:

"First, we study the droplet activation of polydisperse ambient aerosol particles, and then we present a deliquescence experiment with 200 nm Sodium Chloride (NaCl) particles."

We also added some information to the caption of Fig. 5 to clarify which type of particles we used:

"[...] during the two droplet activation experiments conducted with polydisperse particles from ambient laboratory air".

Lines 183-186: It seems odd to have ice formation mentioned here as describing the activation shown in Fig. 5. Would contaminants in the salt be expected to be a major fraction of all CCN? Perhaps I did not understand. But the salient question is why the response is it not a square pulse, if the instrument truly expected to act as a CCN instrument (conjecture made on line 203 that it is possible). I conject that most evidence supports that it cannot act with the resolution of a CCN instrument, struggling to more than define ~water saturation at low temperatures, because conditions across the lamina vary and the aerosols may not stay fully in the lamina. Activation may always be skewed from the true response. Were any experiments done with monodisperse particles? If not, I suggest making it clear when introducing these experiments that they were primarily to show that a liquid activation response starts around 100%.

**Response**: Please see our response to the previous comment: the droplet activation experiments were conducted with polydisperse particles from ambient laboratory air, not with sodium chloride (NaCl). Regarding the last part of the comment, we propose rephrasing L169 to improve clarity:

"The objective of these experiments is to evaluate PINCii's performance by comparing the $RH_{lam}$ measured to values predicted by theory and previous laboratory experiments, both for the deliquescence of NaCl and for the droplet activation of polydisperse ambient particles just above water saturation ($RH_{liq}$=100 %)."

Line 193: This is a very useful experiment, but related to my previous comment, where are the results toward 105%? Perhaps freezing interfered?

**Response**: After deliquescence, we see that the particles continue to grow. However, because of the coarse resolution of the OPC, it is difficult to interpret the data after 95% $RH_{i,lam}$. We also decided to limit the figure to 70-95% $RH_{i,lam}$ since the focus here is the deliquescence occurring around 77% $RH_{i,lam}$.

Lines 198-200: While I can see the OPC resolution issue easily resolved (with extra expense of course), should not the inability to resolve the true lamina RH and its uncertainty at these temperatures be mentioned? This was discussed at length in the Richardson work referenced, as it considered thermal differences on the wall, so could be referenced in this regard.

**Response**: We propose modifying and adding a discussion to the text:

"These results suggest that PINCii can be used to study droplet activation at cold temperatures. Such a feature could potentially be further developed to use PINCii as a low temperature cloud condensation nuclei counter (CCNC). This would however require a more rigorous characterization of the $RH_{lam}$ and its uncertainty. Indeed, the $RH_{lam}$ within the chamber is inferred based on measurements of the chamber wall temperatures and flow rates, and is calculated at the average lamina position using the equations from Rogers (1988). Because the thermodynamic conditions are estimated using numerical routines, it is difficult to calculate the uncertainty in these conditions. Richardson (2009) used the Monte-Carlo methods to perturb and randomly sample analytical solutions to the Rogers (1988) equations in order to explore the uncertainty in the average lamina conditions in more detail. They found the uncertainty of the average lamina temperature to be insensitive to the wall temperatures and operational conditions, yet found quite large uncertainty in the supersaturation values (up to 10 %). Based on this, it is clear that using PINCii to explore low temperature droplet activation processes would require careful development. For example, more experiments with well characterized aerosol particles of known size distribution(s) and chemical composition(s) would be necessary and could be used to provide a reference to better constrain the measurements, as suggested by Richardson (2009). Moreover, it would also likely require using an optical detector with more size channels at the exit of the chamber."

Line 220: Droplet breakthrough concentrations are defined, but what aerosol concentrations entered the instrument?

**Response**: We added the missing information line 218:

"[...]with an aerosol sample consisting of 200 nm ammonium nitrate ($NH_4NO_3$) particles at a concentration of $\approx$350 #/cm$^3$."

Lines 221-224: What is the explanation for the temperature dependence of these results. Others have not found such strong limitations at 103% at -20°C. Is it the long growth time for droplets in the PINCii, meaning that this is an important factor in design of such instruments for measurements in different T ranges? Have you investigated predicted droplet sizes such as by using a microphysical model (e.g., as done for ice)?

**Response**: The temperature dependence of the droplet breakthrough is related to the kinetics of droplet growth and has been observed and reported by others (Chou, 2011; DeMott et al., 2015; Garimella et al., 2016). Concerning the actual RH values obtained from the droplet breakthrough experiments, although others have not found such limitations at -20 °C (Nicolet et al., 2010; Sullivan et al., 2010; DeMott et al., 2015), which might be due to different experimental conditions used (i.e. aerosol particles and size of the aerosol particles used; temperature of the evaporation section, etc.), Garimella et al. (2016) shows very similar results with SPIN. It is possible that the elongated design of PINCii and SPIN allows for longer growth time for droplets in the main chamber, which may reduce the evaporation section's effectiveness despite it also being elongated. As suggested, modeling the droplet growth and evaporation would be required to confirm this hypothesis and if one aimed to conduct experiments close to droplet breakthrough. However, we do not explore this further in this manuscript.

Line 231: Can you be more explicit about what you mean by absent kinetic limitations? It means that you did not explicitly calculate size effects and assumed equilibrium growth? Just clarifying, as this can of course make a large difference.

**Response**: Here we mean that the RH can be considered equivalent to the water activity under thermodynamic equilibrium conditions (Koop et al., 2000). We changed the text line 231:

"[...] used to represent homogeneous freezing onset assuming equilibrium conditions".

Lines 256-257: CFDCs are presumably designed to overcome all but the strongest consumption, depending on aerosol concentration of course.

**Response**: We believe the reviewer is referring to competition for water vapor consumption by the freezing and growing ice crystals. Yes, we agree with the reviewer's comment and, at the conditions operated and these aerosol concentrations, we do not expect a consumption of water vapor that would represent competition for freezing. What we explain is that we use the highest RH conditions in the lamina that represent extreme RH conditions calculated from paired thermocouples. Thus, we identify the highest RH using the highly spatially resolved thermocouple measurements and suggest that these lead to a more representative picture of the RH conditions to which the particles are exposed.

Lines 258-260: How does the high resolution of temperature measurements enable real-time calculation of lamina conditions? Wouldn't you need a fluids model to do that? In real-time? I mean that I don't think one can simply assume that the temperatures at one position define the conditions in the lamina at that position. Without reference to the parcel history before that? I am not a fluids expert, so perhaps I am not correct here.

**Response**: The high spatial resolution of the temperature measurements enables us to identify temperature anomalies along the chamber walls. There is no real-time calculation of the lamina conditions, which would indeed require a real-time fluid mechanics model. We rephrased the sentence to avoid misunderstanding:

"The high spatial resolution of PINCii's temperature monitoring (thermocouples in Fig. 1) enables us to identify chamber-wall temperature anomalies and calculate lamina conditions in more detail when post-processing the experimental data."

Lines 265-267: This is interesting though the use of RH extremes and their calculation is not totally convincing. And can one assume that the Koop line is necessarily the target, given for example the studies of Schneider et al. (2021)?

**Response**: We agree that the Koop line is not necessarily the target. We merely use it here as a base for comparison as it does represent a reasonable and widely accepted approximation of homogeneous ice nucleation. As the referee mentions, previous studies have shown deviation from the Koop et al. (2000) line (Garimella et al. 2016; Welti et al. 2020; Brunner and Kanji 2021; Schneider et al. 2021), which we also observe in our results. Here we use the comparison to the Koop line to propose a new plausible explanation for the observed discrepancies, which is that homogeneous freezing is triggered by the extreme conditions in the lamina. We propose adding the following information to the text lines 260-262, where we also added information concerning the calculation of the RH extremes:

"The method used to identify the most extreme conditions is described in detail in Castarède (2021), and we briefly summarize it here. The idea behind this method is that if a specific location within the chamber favors homogeneous freezing, i.e., conditions above the Koop et al. (2000b) line, any ice crystals formed will continue to grow, even if the remaining aerosol trajectory experiences less favorable conditions for homogeneous freezing, i.e., conditions below the Koop et al. (2000b) line. In practice, the most extreme conditions are determined as follows: first, the lamina conditions ($T_{lam}$ and $RH_{lam}$) are calculated for each pair of opposing thermocouples. Ideally, the next step would be to determine the Euclidean distance between each lamina condition and the Koop et al. (2000b) line. Next, the closest point to the Koop et al. (2000b) line would be identified as the most extreme condition if all data points are below it. If one or more data points are measured above the Koop et al. (2000b) line, the point furthest above it would be considered the most extreme condition. More details can be found in section 3.3.4 and Fig. 3.8 of Castarède (2021). It is important to note that we use the Koop et al. (2000b) line as a reference to compare our results because it is a widely accepted approximation of homogeneous ice nucleation, even though previous studies have shown deviations from this line (Garimella et al. 2016; Welti et al. 2020; Brunner and Kanji 2021; Schneider et al. 2021). Moreover, since the onset of homogeneous freezing is only weakly dependent on temperature compared to saturation conditions, and considering that the uncertainties associated with $T_{lam}$ in the homogeneous freezing regime are low (cf. Fig. 13b), the method described here can be simplified by finding the highest $RH_{i,lam}$ within the lamina, which then represents the most extreme conditions."

Figure 9 and similar plots: At least for me, these require a lot of focus to interpret, as compared to a 2D plot of T vs RH with AF given as contours. I understand it is a preference.

**Response**: We understand that the 3D plots are not as easy to interpret as 2D plots. Considering that both reviewers commented on the 3D plots, we decided to modify Figures 8, 9 and A1 to 2D plots. The updated figures are attached at the end of this document.

**Measurement uncertainties**

Lines 314-316: It seems that uneven spots of wall cooling will be a symptom of any coolant delivery system. It might be distinct for any particular system, but it will have an impact on communicating to the aerosol lamina, which requires further analysis (Richardson, 2009).

**Response**: We thank the reviewer for their comment, and we added the following to the text:

"In addition, the complex combination of PID-controlled heating pads and pulse-injected refrigerant may result in wall temperature fluctuations which also increase uncertainty in the temperature and humidity conditions to which particles are exposed. This is explored in depth in Richardson (2009), who modeled the responses of the thermodynamic variables in the aerosol lamina to different temperatures perturbations (temperature oscillations, gradients, etc.) and investigated the impact of these thermal non-idealities on freezing conditions."

Background discussion: Greatly appreciated this.

- Line 320: CFDCs "typically" need to be warmed, drained, etc. I say that because in aircraft scenarios with long flight hours, complete melting and evacuation is unrealistic, so commonly icing is redone at the same stable icing temperature as used in the first process. This does melt and refreeze the ice surface.

   **Response**: We agree with the referee's comment, and we modified line 324 accordingly:

   "When the ice layers become unstable and increase the background, the CFDCs typically need to be warmed, drained, re-cooled and re-iced to continue experiments."

- Line 323: Can you say something about the volume flow rate of water, or perhaps the volume that is filled and how long the fill occurs? It must matter how fast the air volume is filled. The 1 mm ice thickness is more than I am accustomed to, which is why I ask.

   **Response**: We added the missing information to lines 326-327:

   "Water is then flowed into the chamber via the exit hole (k in Fig. 1) with a flow rate of approximately 6 L.min$^{-1}$, until the water level sensor detects water and stops the filling. This filling typically takes about 1 minute."

- Line 329-331: The procedure of temperature control after icing is a bit unclear. You immediately change the wall temperatures to -5°C after water is introduced, or after it has drained? Then step-wise cooling to the operating conditions? Both walls? I suspect that many have their own procedures for achieving low backgrounds, so great to understand this clearly to be tested against others.

   **Response**: The wall temperature is changed to -5 °C during the draining, then both wall temperatures are decreased to the operating conditions in a stepwise manner. We rephrased L329-331 to improve clarity:

   "To avoid cooling too rapidly, the setpoint temperatures are changed from -23 and -20 °C to -5 °C while the water is being drained out. Then both wall temperatures are decreased stepwise by 5 °C increments, where the temperature is stabilized at each interval until the desired experimental conditions are reached."

- Line 334: The outlet is dried. How is that done? Overpressure of air before connecting OPC?

   **Response**: We added the missing information to the text:

   "When the desired experimental conditions are achieved, the outlet (which tends to accumulate water) is dried before the OPC is re-attached. This is typically done using lint-free laboratory tissue or gently blowing pure nitrogen at the outlet while keeping the outlet valve closed."

- Line 339: Is a progressive cooling direction of sampling important for limiting background? This is also common with other CFDCs, so would be good to say.

   **Response**: Yes, we have observed that doing ramps from the warmest to the coldest temperatures tends to help preserve a good and stable background, as it was observed with other CFDCs. At warmer

temperatures, the absolute humidity is higher, and the diffusion is faster, so frost might develop faster and lead to a poor background. We added this information to the text line 339:

"It is important to note that experiments are typically done with ramps from the warmest to the coldest temperatures, as we have observed that doing so helps preserve a good background."

- Line 345: It is great to be open and honest about this, but I hope you are speaking for PINCii users primarily. There are no wider CFDC community users I know who are not practically aware that ice conditions evolve. This is especially important for heterogeneous freezing experiments. It might be great to explore the different knowledge bases out there amongst CFDC users in a workshop scenario.

   **Response**: We agree with the referee's comment. Our statement line 345 was merely to try to be open and honest with future PINCii users.

Lines 365-366: Can you explain this statement better? "On the other hand, the colder temperatures are within the realm of homogeneous freezing, and thus the link between INP and observed ice above water saturation is ill-defined." It is quite possible for water saturation to be exceeded to stimulate ice nucleation at low temperatures if a particle is sufficiently small and hydrophobic, no? Not every particle will freeze at the homogeneous freezing condition for solutes.

**Response**: Here we simply mean to say that at temperatures below homogeneous freezing, we do not need to distinguish between liquid droplets and ice crystals because any formed ice crystals at $RH < RH_{hom}$ would grow large enough to be distinguished from liquid droplets. And if the freezing occurs at $RH > RH_{hom}$, then it would be impossible to tell the difference between heterogeneous and homogeneous freezing unless a very well-defined aerosol sample was being used. For instance, in field measurements this distinction would not be possible. We propose rephrasing line 365-366 to clarify:

"On the other hand, the colder temperatures are within the realm of homogeneous freezing, and thus the distinction between INPs freezing and ice observed above the RH conditions for homogeneous freezing cannot be defined. An exception to this would if laboratory experiments were done with an aerosol sample known not to freeze homogeneously. Then any ice observed above the RH conditions for homogeneous freezing could be attributed to INPs."

Lines 401-404: I feel that for operations in the mixed phase regime, this might need more analysis. I understand the focus here is on lower temperatures.

**Response**: We agree with the referee that this could be further investigated for operations in the mixed-phase regime, although we do not explore this here.

Line 411: "inside the chamber" or along the chamber walls? You are not really modeling what is happening in the lamina are you? That is, the calculation is analytical.

**Response**: It is indeed the temperature distribution along the chamber walls. We modified the sentence in line 411 to avoid confusion.

Lines 415-417: I see the statement at the end of this paragraph that these calculations do not represent all factors in knowing the RHi uncertainty, but the values stated here based on (I think static, point-to-point measurements) seem extremely low for these low temperature conditions, low even if you were referencing RHw uncertainty. Is the uncertainty in the T measurement itself also factored in? I may be wrong, but I feel that this is not a solvable

analytical problem and really requires an approach such as the Monte Carlo approach taken in Richardson (2009), as stated in Richardson et al. (2010).

**Response**: The results presented in Figure 13 only take into account the spatial variations in the temperature and RH, and not the uncertainty of the temperature measurement itself. We do expect that the actual measurement uncertainties will be higher and highly dependent on the operating mode. Note also that what we present here is the standard error and not the standard deviation, which would be higher. We propose rephrasing lines 415-417 to clarify what is accounted for in Figure 13:

"In Fig. 13, the standard errors of $T_{lam}$ and $RH_{i,lam}$ calculated from wall temperature profiles measured over the full operational range of PINCii are presented. The standard errors were calculated using temperature measurements made during the previously presented experiments of droplet nucleation (Fig. 5), homogeneous freezing (Figs. 8 & A1) and particle loss (Fig. 11). The $T_{lam}$ and $RH_{i,lam}$ were calculated for each temperature measurement along the chamber walls before calculating the standard errors of the obtained values. The standard errors were then inserted into an empty $RH_{i,lam}$, $T_{lam}$ coordinate matrix, and missing points were calculated using a nearest neighbors linear interpolation."

And lines 423-426:

"Note that the uncertainties presented here only account for the spatial variations of the temperature measured by the thermocouples distributed along the chamber walls and do not account for other measurement uncertainties (e.g., thermocouple uncertainty or uncertainty related to flow rate, pressure etc.). Thus, the results presented here represent ideal (minimum) uncertainties. The actual measurement uncertainties are higher, highly dependent on the operating mode used, and their calculations are not easy to generalize. Although such calculations are not further explored in this work, some methods have been developed, such as the Monte Carlo approach taken in Richardson (2009)."

**Conclusions**

Lines 442-445: Should you not be able to compare conditions for significant freezing ala a time dependent freezing calculation using Koop et al, say where 1% freezing is expected? I did not understand the significance of the "onset" condition. One might want to look for agreement at a range of freezing fractions for the residence time of the chamber. Why only focus on onset conditions?

**Response**: We agree with the referee that, in principle, the Koop et al. (2000b) theory and the PINCii residence time, could be used for time-dependent freezing calculations (e.g., fitting a nucleation rate to achieve a matching activated fraction) to provide another benchmark for CFDC results. However, here we prefer using the freezing onset (defined as the inflection point in the slope(s) of the activation curve(s)) as it indicates the conditions when the measured aerosol particles start producing elevated amounts of ice crystals. We believe that this condition indicates that the rate of ice nucleation is fast relative to the observational time scale, and we find this approach to be more useful compared to, for example, selecting an arbitrary activated fraction value like 1 %. Using the residence time and a nucleation rate constant to extract activated fractions through the Koop theory might be problematic given that the user does not know exactly where (when) the nucleation takes place within a CFDC. The reviewer's suggestion that CFDCs (including PINCii) could be used to explore the regime shift between time-dependent freezing and more deterministic-like freezing is a good one and would be an excellent area for further exploration. However, it goes a bit beyond the intentions of this manuscript, which is to provide a foundation for using PINCii and interpreting its measurements.

Line 446: I think readers would like to understand better how the Rogers (1988) model was modified. Are there new equations to predict the temperature at the interface of the thicker ice wall and the interior of the chamber? That is, what beyond altering the lamina distance is involved? How could others take advantage of this?

**Response**: We understand that some more information concerning the modified equations would be beneficial. As this was explained in depth in section 3.3.2 of Castarède (2021), we propose mentioning the thesis in the main text in case readers would be interested in more details. We also added more explanation to the modification made and their implications (see comment on section 2.4 at the beginning of this document).

Figure A3: I am curious if this figure suggests detection limit issues in dependence on aerosol size at temperatures below -50°C, if a threshold size is used for ice detection? Is ice detection size variable for low T studies?

**Response**: Figure A3 suggests that with the finite residence time inside PINCii (or any other CFDC), the kinetics of ice crystal growth limits what can be detected as ice if phase discrimination at the exit of the chamber is purely based on size. This was also discussed in Welti et al. (2020) and Burkert-Kohn et al., (2017). Based on this, different threshold sizes (e.g., 1 or 2 µm) or lower flow rates could be used for studies at low temperatures. Please see also our earlier responses related to the detection size cutoffs, where we have tried to make clear that the choice of detector(s) and cutoff should be considered in the context of individual experiments or intended measurements.

**Editorial notes**

Line 79: Mobile sounds like something that propels itself, although I realize it has been used in this field for devices that can be taken outside the laboratory. Transportable?

**Response:** We changed "mobile" to "transportable" as suggested.

Line 114: May not need "reproducible" here since the sentence ends with "reproduced."

**Response**: We agree with the referee and removed "reproducible" line 114.

Line 116: "most other" existing chambers

**Response**: We have rephrasing line 116 according to the referee's comment on line 122 (see above):

"Compared to all but a few CFDCs, and in contrast to the first PINC, PINCii has an elongated design [...]"

Line 207: Suggest that it would be helpful to add at the end of this first sentence "to reduce droplets to unactivated sizes."

**Response**: We thank the referee for this suggestion. We modified the text line 207:

"Droplet breakthrough refers to chamber conditions at which the evaporation section no longer functions effectively to evaporate droplets to sizes where they can be distinguished from ice crystals".

Line 430: "most "previous generations of deployable CFDCs…"

**Response**: We modified the text line 430 as suggested:

"The upgraded capabilities of PINCii relative to most previous generations of deployable CFDCs are highlighted."

Line 462-463: Suggest to remove "while remaining transportable." Most any CFDC is transportable.

**Response**: We agree with the referee, and we removed "while remaining transportable" lines 462-463.

Line 465: Line 446: Suggestion to end sentence with "At temperature and RH conditions for which the ambient INP concentration exceeds its limit of detection."

**Response**: We propose rephrasing as follows:

"This means that PINCii is suited for sampling low INP concentrations (< 10 #/L)."

**References** (not already in paper)

Bi, K., G. R. McMeeking, D. Ding, E. J. T. Levin, P. J. DeMott, D. Zhao, F. Wang, Q. Liu, P. Tian, X. Ma, Y. Chen, M. Huang, H. Zhang, T. Gordon, and P. Chen, 2019: Measurements of ice nucleating particles in Beijing, China. Journal of Geophysical Researh: Atmospheres, **124**, 8065–8075, https://doi.org/10.1029/2019JD030609.

DeMott, P. J., K. Sassen, M. Poellot, D. Baumgardner, D. C. Rogers, S. Brooks, A. J. Prenni, and S. M. Kreidenweis, 2003: African dust aerosols as atmospheric ice nuclei. Geophysical Res. Lett., **30**, No. 14, 1732, doi:10.1029/2003GL017410.

DeMott, P. J., M. D. Petters, A. J. Prenni, C. M. Carrico, S. M. Kreidenweis, J. L. Collett, Jr., and H. Moosmüller, 2009: Ice nucleation behavior of biomass combustion particles at cirrus temperatures, Journal of Geophysical Researh: Atmospheres,, **114**, D16205, doi:10.1029/2009JD012036.

Kasparoglu, S., Perkins, R., Ziemann, P. J., DeMott, P. J., Kreidenweis, S. M., Finewax, Z., Deming, B. L., DeVault, M. P. and Petters, M. D. (2022). Experimental determination of the relationship between organic aerosol viscosity and ice nucleation at upper free tropospheric conditions. Journal of Geophysical Researh: Atmospheres,, **12**7, e2021JD036296. https://doi.org/10.1029/2021JD036296.

Schneider, J., Höhler, K., Wagner, R., Saathoff, H., Schnaiter, M., Schorr, T., Steinke, I., Benz, S., Baumgartner, M., Rolf, C., Krämer, M., Leisner, T. and Möhler, O.: High homogeneous freezing onsets of sulfuric acid aerosol at cirrus temperatures, Atmos. Chem. Phys., 21(18), 14403–14425, doi:10.5194/acp-21-14403-2021, 2021.

**Updated figures:**

[Figure]

Figure 8. Homogeneous freezing of 200 nm NaCl particles plotted as a function of $T_{lam}$ and $RH_{i,lam}$ represented in three different manners: (a) AF of aerosols with $d_p \geq 5$ µm plotted as a function of the average $T_{lam}$ and $RH_{i,lam}$. (b) AF of aerosols with $d_p \geq 3$ µm plotted as a function of the average $T_{lam}$ and $RH_{i,lam}$. (c) AF of aerosols with $d_p \geq 3$ µm plotted as a function of the average $T_{lam}$ and the maximum $RH_{i,lam}$. The theoretical curves for water saturation (solid black line) and homogeneous freezing (dashed black line, calculated with $\Delta aw = 0.2946$ following Koop et al., 2000b) were added for supplementary information. In each plot, the color scale is used to represent changes in

the AF. The white region in the color bars represents the ice nucleation onset, which was estimated using the median of the inflection points obtained for each activation curve (see A1 for more information).

[Figure]

Figure 9. Heterogeneous freezing of size-selected 200 nm NX-illite particles. The AF of aerosols with $d_p \geq 3$ μm is plotted as a function of the average Tlam and $RH_{i,lam}$. The AF= 1% is plotted as circles for comparison with the AF=1 % onset of similar particles reported in Welti et al. (2009), shown here as black squares.

[Figure]

Figure A1. Homogeneous freezing of 200 nm particles generated from a natural salt sample collected in the Qaidam basin, China. Note that, due to an instrumental malfunction, the CPC was not running and thus the AF could not be calculated for this experiment. The concentration of aerosols with $d_p \geq 3$ µm is plotted as a function of the average Tlam and the maximum $RH_{i,lam}$, and the color scale is used to represent changes in the concentration. The white region in the color bars represents the ice nucleation onset which was estimated using the median of the inflection points obtained for each activation curve (see A1 for more information). The theoretical curves for water saturation (solid black line) and homogeneous freezing (dashed black line, calculated with $\Delta aw = 0.2946$ following Koop et al., 2000b) were added for supplementary information.

---

## Author Comment (AC3)

**Referee comment: Development and characterization of the Portable Ice Nucleation Chamber 2 (PINCii), Castarède et al., 2023**

The manuscript presents a new instrument to measure ice-nucleating particle concentrations, both in the laboratory and the field. Even though there is already a high number of instruments of the same type, PINCii is a further improvement of the measurement technique. Furthermore, the authors discuss a new approach to analyze the data of a CFDC. To validate the instrument, multiple experiments including deliquescence, homogeneous freezing and heterogeneous freezing were performed and the results are compared to literature values from different studies.

Overall, the manuscript is well written and structured and gives detailed explanations of the work that has been done. However, there are some points, listed below, that would further improve the quality of the paper. My comments are structured in first more general comments and second minor comments.

Response: We thank the referee for their useful comments which helped improve the manuscript. We include our responses below, in the context of individual comments.

**General comments**

- Given the length of the text, there is a quite high number of figures, such as e.g. in section 2.4. I would recommend to reduce the number of figures and instead explain the outcome of the measurements in more detail in the text. The figures can then be either deleted or moved to the appendix.

Response: We understand the reviewer's comment and we combined Figures 3 and 4 into one figure. We attach the updated figure at the end of this document. We believe that the rest of the figures are necessary to the manuscript, and we would like to keep them in the main text.

- Figure 8, 9 and A1 are very complex and hard to understand. They should be simplified to a 2D plot, with AF as a color scale. If the change in AF is the important parameter for defining the onset, then the color code can also be presented as Δ AF.

Response: We understand that the 3D plots are quite complex and hard to interpret. Since this issue was brought up by two reviewers, we decided to modify Figures 8, 9 and A1 to 2D plots. We attach the updated figures at the end of this document.

- How much does a potential underestimation of the number of ice crystals due to the binning of the OPC and the set ice threshold contribute to the total uncertainty of the INP concentration? The authors provide a calculation of the ice crystal growth, however, this assumes a spherical ice crystal and a constant mass accommodation coefficient of 0.3. Based on this calculation, the threshold for ice crystal detection in the OPC was set. However, since the binning of the OPC is quite broad, some particles might not be counted as ice (or droplets), because they are not detected in the respective channel of the OPC. This could be especially relevant when measuring INPs in low concentrations.

Response: Thank you for your comment. We first want to clarify that the ice crystal growth calculations were made retroactively to ensure that our observations are aligned with theoretical expectations, and not to set the threshold for ice crystal detection in the OPC. The choice of both detector and cut-off size should be determined based on the experimental parameters.

Concerning the potential underestimation of the number of ice crystals detected, there is no single

uncertainty that can be quoted in this case. We believe that when the chamber is operated to capture clear ice nucleation onsets, the uncertainties are likely small or negligible compared to the natural variability of the INP concentrations (for ambient measurements). The ability to change the ice crystal size detection threshold allows us to avoid undercounting ice crystals because the detection threshold can be chosen as a smaller (or larger) size to account for the slower (or faster) growth of the ice crystals depending on operating temperature(s). Figure A3 suggests that at -21 ℃, an ice crystal grows to 5 μm in diameter at $RH_i$ = 104 % within the 15 s residence time of PINCii. This implies that, at -21 ℃, using a 5 μm detection threshold will not yield any reported INPs for $RH_i$ < 104 %. However, this is not of concern because INPs do not freeze at such low RH for this temperature range where immersion freezing would dominate, and RHw would need to be at 100 % or higher, thus far exceeding 104 % $RH_i$. In theory, one could define that PINCii is not suited for measurements at $RH_i$ < 104 % for -21 ℃, i.e. that a limit of operation can be defined rather than an uncertainty. But this will not affect the applicability of PINCii since immersion freezing at $RH_i$ =104 % would be thermodynamically impossible.

Similarly, for colder temperatures, the threshold can be reduced to 3 μm. For instance, from Fig. A3 at -31 ℃ and $RH_i$ = 104 %, an ice crystal grows to 3 μm. Here again, any ice nucleation occurring at $RH_i$ < 104 % would not be detected. As such a limit of operation can be defined but also immersion freezing would $RH_i$ >>104 % and thus would not be of concern. One could argue that already at 105 % at -35 ℃, deposition nucleation is reported (Welti et al., 2009 ACP). In this case, an interference from liquid droplets is not a concern, so the ice detection threshold can be further decreased to 2 μm or even 1 μm if the aerosol being sampled is below 800 nm in diameter so as to not interfere with the 1 μm OPC channel.

Therefore, the flexibility in choosing the ice threshold is key to avoid undercounting ice crystals. Moreover, in the absence of such flexibility, one can define a RH range limit of operation such as: above RHi >104 % and below water droplet breakthrough.

In theory, this exercise is also dependent on the aerosol particle size being sampled. If super micron particles are being sampled in the field, one has to define the ice threshold as that above the largest sampled unactivated aerosol signal in the OPC. And a choice in the ice crystal threshold size can always be customized to avoid undercounting and to define a limit of operation. So the colder temperature measurements can even use a size threshold of 1 μm as long as the sampled aerosol sizes are below 500-800 nm and are not detected in the optical 1 μm channel.

We also note that the calculations in Figure A3 assume spherical ice crystals and compact spheres. However, this is not a poor assumption for detection sizes of below 5 μm as non-spherical features at such small sizes are hard to detect or quantify. Moreover, even in depolarization detectors, crystals of this size appear highly spherical (Mahrt et al., 2019 AMT).

**Minor comments**

- L6: The phrase "very low concentrations" should be supported with numbers of the range of the limit of detection

**Response**: We agree with the reviewers, and we have rephrased the sentence to add some numbers:

"Notably, a specific icing procedure results in low background particle counts, which demonstrates the potential for PINCii to measure INPs at low concentrations (< 10 #/L)."

- L21-22: In line 21 you are writing "heterogeneous nucleation", however, in the following sentence in L22 you call it "heterogeneous ice nucleation". You should stick to one term, preferably the second one.

**Response**: We agree, and we changed to "heterogeneous ice nucleation".

- L39: Mention that the CFDC-IAS has a cylindrical shape

**Response**: We thank the reviewer for this suggestion. We added a mention to the shape of the CFDC-IAS L39:

"[...] the cylindrical CFDC-IAS (Handix Scientific, Boulder, Colorado, USA) [...]'

- Figure 1: A list inside the figure explaining the letters (a) to (k) would help for an easier understanding. The color of (f) (refrigerant cooling coil pipes) should be changed, because it is difficult to differentiate it from the other items.

**Response**: As the figure is already quite full, we believe that adding text next to it would make it harder to read. Moreover, the annotations are explained in the caption located directly under the figure. Considering this and the fact that the figure was optimized for the one-column format, we would rather keep the figure as is. We agree with the reviewer that the color of the cooling pipes was difficult to see, and we changed the color as suggested (see updated figure at the end of this document).

- L89: Briefly explain what ETH-IODE is and what it is used for

**Response**: We propose to rephrase the sentence at L89 to:

"[...] designed for mounting the Ice Optical DEpolarization detector (IODE; Nicolet et al. 2010) used to distinguish between water droplets and ice crystals."

- L101: Explain the abbreviation R23

**Response**: R-23 is the official product name of this refrigerant and is not necessarily an abbreviation. Nonetheless, we propose rephrasing L101 to clarify:

"[...] cold injection points given that the refrigerant R-23 (trifluoromethane ) operates with [...]"

- L117: How much longer is the main chamber of PINCii compared to other CFDCs? You should give at least a range of numbers

**Response**: Thank you for this suggestion. We rephrased L117 to indicate the chamber lengths:

"[...] PINCii has an elongated design, where the main chamber (100 cm) and the evaporation section (43 cm) are longer than in other instruments (e.g. PINC and SPIN have main chambers of 56.8 and 100 cm, and evaporation sections of 23 and 25 cm respectively)."

- Figure 5: The symbols of the first and second experiment are very hard to distinguish in the plot. As it is written in the text, the data are presented as normalized values, so it might not be needed to present them in different symbols. If the authors think, that it gives the reader some value to know which data points were recorded on which day, they should divide the figure in two sub plots. It might be also beneficial to mark the range in which the data points represent either an activated cloud droplet or an ice crystal.

**Response**: In Figure 5 we want to show that the data points overlap since it means that the results are identical for both experiments (after normalization) even though they were done on two separate days; thus, making them distinguishable is not our objective. However, since there were indeed two experiments on two different days, we do want to present the data with different symbols.

The experiments presented in Figure 5 were conducted with a temperature gradient along the evaporation section (so the evaporation section does not evaporate anymore). Thus, we cannot distinguish activated cloud droplets from ice crystals. We modified the text to make it more clear:

L167 "In this work, we use this feature to show that the chamber can actively grow droplets."

And added a sentence L169:

"First, we study the activation of polydisperse ambient aerosol particles, and then we present a deliquescence experiment with 200 nm Sodium Chloride (NaCl) particles. For both experiment types, the temperature gradient along the main chamber is extended to the evaporation section, so the evaporation section is no longer evaporating droplets."

- L234-239: The authors should elaborate a bit more the outcome of figure 8(a) by giving numbers e.g. at which RH_lam activation happens for different temperatures and how much it differs from the Koop line.

**Response**: We added more information L234-239 and we hope that changing figure 8 to 2D plots will also improve clarity.

"Presenting data in this way shows an increasing deviation from the Koop et al. (2000b) curve towards lower $RH_{i,lam}$ as the temperature increases from $T_{lam}$ = -45 °C, with the maximum deviation at $T_{lam}$ =~-40 °C where the freezing onset is observed 7.2 % $RH_{i,lam}$ below the Koop et al. (2000b) curve. Figure 8a also shows a slight deviation from the Koop et al. (2000b) curve at $T_{lam}$ = -49 °C, where the freezing onset is observed 1.7 % $RH_{i,lam}$ above the Koop et al. (2000b) curve. [...]"

- L247-248: How is it seen that some ice crystals did not grow to 5μm. I guess it can be seen by the ice threshold that is shifting to a lower RH_lam from Fig. 8(a) to 8(b). However, a short note on that might be helpful for the reader.

**Response**: Yes indeed, it is seen by the ice nucleation onset that is shifting towards lower $RH_{i,lam}$. We added a note L247-248 to be more clear:

"Although the change in the size threshold does not affect the activation curves for $T_{lam}$ > -45 °C, the ice nucleation onsets shift towards lower $RH_{i,lam}$ for Tlam < -48 °C, illustrating the existence of activated crystals that have not grown fully to 5 μm for these temperatures. "

- L274-276: Replace one of the "significantly"

**Response**: The "significantly" L276 was changed to "substantially".

- Figure 9: In the caption, replace "triangles" by "squares"

**Response**: Thank you for spotting this mistake, we changed it to "squares".

- L329: Add a short note why the rapid cooling should be avoided

**Response**: We rephrased L329 to add more information:

"To avoid cooling too rapidly, which would deteriorate the quality of the ice-coating and lead to high background counts, the setpoint temperatures are changed [...]"

- L337: "exceptionally" and "mediocre" need to be defined in terms of values

**Response**: The values are given a few lines later (L341-343). The sentence L337 merely introduces Figure 10.

- L341: Remove "exceptionally"

**Response**: We removed "exceptionally" as suggested (also L337).

- L343: A short discussion about the background concentration after 3 ramps is missing

**Response**: We propose to add some information L343:

"However, after three $RH_{i,lam}$ ramps, the ice layer clearly deteriorated and the median background count is 60.0 #/L, which is too high for continuing with the measurements."

- Figure 11: Was the droplet break through only measured for four of the scans?

**Response**: Yes, as presented in Figure 7, droplet breakthrough was measured for four temperatures (-20, -25, -30 and -35 ℃).

- L465: Rephrase to "sampling from sources with INP concentrations as low as …"

**Response**: We rephrased as suggested:

"This means that PINCii is suited for sampling low INP concentrations (< 10 #/L)."

- Figure A1: This figure is mentioned quite often in the text. Therefore, I suggest to move it from the appendix to the main text.

**Response**: Figure A1 is mentioned only twice in the text (L272 and L415) and therefore we would like to keep it in the appendix. There is however the issue that both the first figure and the first section of the appendix have the same name (A1), which might be confusing when mentioning one or the other. This is a typesetting issue which we expect will be dealt with in the case of acceptance.

**Updated figures:**

[Figure]

**Figure 1**. Schematic of PINCii and its cooling system, including the main elements of the chamber: (a) the sample inlet, (b) the sheath flow inlets, (c) the water level sensor port, (d), (e) and (h) the SustaPEEK flanges thermally isolating sections of the chamber, (f) refrigerant cooling coil pipes, (g) support bars, (i) window port, (j) the lower evaporation section with material removed, and (k) the exit hole. The location of the coolant injections, heating pads and thermocouples on a single wall are also depicted. The injection of coolant to the different chamber sections is controlled independently, while multiple thin capillaries located after the solenoid valves distribute the coolant evenly to the selected section.

[Figure]

**Figure 3**. (a) Flow velocity as a function of ice thickness for fixed wall temperatures of -40.0 and -56.5 °C, chosen to represent homogeneous freezing conditions PINCii's main chamber. These conditions ($T_{lam}$ = -51.3 °C and $RH_{i,lam}$ = 155.7 % at ice thickness = 0 mm, and $T_{lam}$ = -50.1 °C and $RH_{i,lam}$ = 154.7 % at ice thickness = 1 mm) are representative of extreme chamber operations for PINCii, with the greatest potential for buoyancy effects. The lamina position is depicted by the dashed black lines. The white contour line corresponds to a velocity of 0 cm/s and emphasizes where the region with negative velocity starts. (b) Achievable $T_{lam}$ and $RH_{i,lam}$ assuming fixed wall temperatures between -5 and -60 °C and accounting for the droplet breakthrough results presented in section 3.1. The color map represents the reversed flow fraction defined as the ratio between the reverse (upward) flow and the normal (downward) flow in the chamber, assuming a 1 mm ice layer on each wall. The black solid line represents water saturation ($RH_{liq,lam}$=100 %).

[Figure]

**Figure 8**. Homogeneous freezing of 200 nm NaCl particles plotted as a function of $T_{lam}$ and $RH_{i,lam}$ represented in three different manners: (a) AF of aerosols with $dp \geq 5$ μm plotted as a function of the average $T_{lam}$ and $RH_{i,lam}$. (b) AF of aerosols with $dp \geq 3$ μm plotted as a function of the average $T_{lam}$ and $RH_{i,lam}$. (c) AF of aerosols with $dp \geq 3$ μm plotted as a function of the average $T_{lam}$ and the maximum $RH_{i,lam}$. The theoretical curves for water saturation (solid black line) and homogeneous freezing (dashed black line, calculated with $\Delta aw = 0.2946$ following Koop et al., 2000b) were added for supplementary information. In each plot, the color scale is used to represent changes in the AF. The white region in the color bars represents the ice nucleation onset, which was estimated using the median of the inflection points obtained for each activation curve (see A1 for more information).

[Figure]

**Figure 9**. Heterogeneous freezing of size-selected 200 nm NX-illite particles. The AF of aerosols with dp ≥ 3 μm is plotted as a function of the average $T_{lam}$ and $RH_{i,lam}$. The white region in the color bars represents AF=1 %. The AF=1% is also plotted as circles for comparison with the AF=1 % onset of similar particles reported in Welti et al. (2009), shown here as black squares.

[Figure]

**Figure A1**. Homogeneous freezing of 200 nm particles generated from a natural salt sample collected in the Qaidam basin, China. Note that, due to an instrumental malfunction, the CPC was not running and thus the AF could not be calculated for this experiment. The concentration of aerosols with dp ≥ 3 μm is plotted as a function of the average $T_{lam}$ and the maximum $RH_{i,lam}$, and the color scale is used to represent changes in the concentration. The white region in the color bars represents the ice nucleation onset which was estimated using the median of the inflection points obtained for each activation curve (see A1 for more information). The theoretical curves for water saturation (solid black line) and homogeneous freezing (dashed black line, calculated with Δaw = 0.2946 following Koop et al., 2000b) were added for supplementary information.

---

## Referee Report (RR1)

**Referee report: Development and characterization of the Portable Ice Nucleation Chamber 2 (PINCii), Castarède et al., 2023**

After reviewing the author's responses to the referee comments I would recommend the presented manuscript for final publication. The authors replied in a sufficient way, changed text and figures where it was suggested and by that improved the quality of the manuscript.